# The ER anchoring and abundance of the EEF1B complex is affected by tissue-specific alternative EEF1D splicing

Muhammad Jamous[1,*] , Masaki Hosogane[1,2,*] , Xiaoxin Huang[1,3,*] , Mikiko Suzuki[4,5] , Atsushi Hatano[6], Yuichi Shichino[7,8], Shintaro Iwasaki[7,9] , Kazutaka Murayama[10], Masaki Matsumoto[6], Keiko Nakayama[1,11]

**The EEF1B complex plays a central role in translation elongation by reactivating EEF1A for the delivery of aminoacyl-tRNAs to the ribosome. Among its components, EEF1D undergoes alternative splicing to produce one long and several short isoforms, each with distinct N-terminal domains and tissue-specific expression patterns. Although the short isoforms are broadly expressed, their physiological functions remain poorly characterized. In this study, we show that short EEF1D isoforms containing exon 5 interact with the ER-resident scaffold protein KTN1 and RRBP1, thereby anchoring the EEF1B complex to the ER. Mass spectrometry analyses of FLAG-tagged EEF1D identified these interactions, and deletion of exon 5 disrupted ER anchoring, resulting in diffuse cytoplasmic localization of the EEF1B complex. In exon 5 KO mice, this altered localization was accompanied by a reduction in EEF1B subunit abundance in multiple tissues, including the liver, although global protein synthesis rates remained unchanged. Together, these findings uncover an ER-anchoring mechanism controlled by alternative splicing that shapes the spatial organization and abundance of the elongation machinery in vivo.**

## Introduction

Eukaryotic translation elongation factor 1 delta (EEF1D), also known as eEF1Bδ (see Table 1 for alternative nomenclature), is a critical component of the translation elongation machinery. EEF1D functions as a guanine nucleotide exchange factor (GEF) subunit in the eukaryotic elongation factor 1B (EEF1B) complex, which also includes EEF1B2, EEF1G, and Valyl-tRNA synthetase (VARS) (1, 2, 3). The EEF1A protein, in conjunction with the EEF1B complex, forms the larger eukaryotic translation elongation factor 1 (EEF1) complex. The EEF1 complex primarily facilitates the reactivation of EEF1A by catalyzing the exchange of GDP for GTP. Activated EEF1A with GTP delivers aminoacyl-tRNAs to the A-site of the ribosome for translation, thereby contributing to the efficiency and fidelity of protein synthesis (4, 5).

EEF1D is an animal-specific subunit of the EEF1B complex, suggesting a specific role across the animal kingdom (2). Although EEF1D and EEF1B2 share a conserved C-terminal GEF domain, these two GEF proteins are not redundant. In humans, homozygous mutations affecting the GEF domain of EEF1D have been linked to severe clinical phenotypes, including intellectual disability, seizures, failure to thrive, and recurrent aspiration pneumonia (6). Furthermore, studies in mice have demonstrated that homozygous KO of *Eef1d* results in embryonic lethality (7). These findings highlight the critical and non-compensable role of EEF1D in animal species. Compared with EEF1B2, EEF1D is longer and structurally more complex, with functions extending beyond its canonical role as a GEF. EEF1D contains a leucine zipper (LZ) domain within its N-terminal region that facilitates dimerization (2, 8). The N-terminal region also harbors an EEF1G-binding domain that mediates interaction with EEF1G and is required for stable assembly of the EEF1B complex (8). In addition to its role in the EEF1B complex, EEF1D directly interacts with translationally controlled tumor protein through the central acidic region, thereby regulating its GEF activity (9), and with kinectin 1 (KTN1), an ER membrane scaffold protein (10). The C-terminal region of KTN1 is critical for the interaction with EEF1D, suggesting a role for EEF1D in ER-associated processes, including protein synthesis, although the precise binding site within EEF1D has not been defined.

Alternative splicing programs are often cell type—and developmental stage—specific, and they can reconfigure protein–protein interactions, alter domain composition, and

[1]Division of Cell Proliferation, Graduate School of Medicine, Tohoku University, Sendai, Japan    [2]Department of Medical Genome Science, Dokkyo Medical University, Mibu, Japan    [3]Graduate School of Life Sciences, Tohoku University, Sendai, Japan    [4]Center for Radioisotope Sciences, Graduate School of Medicine, Tohoku University, Sendai, Japan    [5]Advanced Research Center for Innovations in Next-Generation Medicine (INGEM), Tohoku University, Sendai, Japan    [6]Department of Omics and Systems Biology, Niigata University, Niigata, Japan    [7]RNA Systems Biochemistry Laboratory, Pioneering Research Institute, RIKEN, Wako, Japan    [8]Department of RNA Biochemistry, Institute of Medicine, University of Tsukuba, Tsukuba, Japan    [9]Department of Computational Biology and Medical Sciences, Graduate School of Frontier Sciences, The University of Tokyo, Kashiwa, Japan    [10]Division of Biomedical Measurements and Diagnostics, Graduate School of Biomedical Engineering, Tohoku University, Sendai, Japan    [11]National Institutes for Quantum Science and Technology, Chiba, Japan

Correspondence: m-hosogane@dokkyomed.ac.jp; nakayama.keiko@qst.go.jp
*Muhammad Jamous, Masaki Hosogane, and Xiaoxin Huang contributed equally to this work

**Table 1. Alternative nomenclature of the EEF1B complex.**

| Gene symbol | eEF nomenclature | Alternative names |
|---|---|---|
| EEF1D | eEF1Bδ | EF-1δ |
| EEF1B2 | eEF1Bα | EF-1β |
| EEF1G | eEF1Bγ | EF-1γ |
| EEF1A1 | eEF1A1 | EF-1α |
| EEF1A2 | eEF1A2 | EF-1α2 |
| VARS1/VARS | Valyl-tRNA synthetase (cytoplasmic) | ValRS |

dictate subcellular targeting (11). Among the EEF1B complex components, EEF1D exhibits functional diversity conferred by alternative splicing (12). In mice, the *Eef1d* gene generates multiple transcript variants, including a long isoform and several shorter isoforms that differ in their N-terminal domains. The longer isoform (EEF1D-L, also known as eEF1BδL) results from the inclusion of a tissue-specific N-terminal exon that encodes a nuclear localization signal and is predominantly expressed in the brain and testis (13, 14). Although EEF1D-L contains a GEF domain, it appears to function primarily as a stress-responsive transcription factor rather than as a canonical GEF (15). Mutations affecting EEF1D-L–specific residues have been reported in individuals with intellectual disability, indicating its importance in neurological function (16). The shorter variants retain the conserved GEF domain, but they have different alternative exons within the N-terminal part between EEF1G (1–43 aa) and LZ (80–115 aa) domains. AlphaFold-based structural prediction suggests that these variants adopt distinct α-helical conformation, implying potential functional differences among isoforms. However, the specific contributions of these short isoforms, as well as their precise tissue expression patterns, remain largely uncharacterized.

This study shows that EEF1D isoforms exhibit tissue-specific expression patterns and identifies specific binding partners of each splicing isoform. EEF1D exon 5 serves as a key determinant for interactions with ER membrane proteins, including KTN1 and RRBP1. Loss of exon 5 results in dissociation of the EEF1B complex from the ER membrane and its redistribution into the cytoplasm in both cultured cells and mouse tissues. Notably, in exon 5 KO mice, we observed a significant decrease in the entire EEF1B complex proteins in a tissue-specific manner, suggesting that exon 5–dependent interactions govern the spatial organization and abundance of the EEF1B complex.

## Results

### Tissue-specific expression and proteomic identification of isoform-specific protein interactions

To investigate the tissue-specific expression of the EEF1D protein, we analyzed lysates from mouse brain, heart, liver, and skeletal muscle (Fig 1A). Phosphatase treatment was used to distinguish phosphorylated from non-phosphorylated forms, as EEF1D is a known substrate for both CDK1 and CK2 kinases (17, 18, 19). In untreated samples, signals with multiple molecular weights were observed; however, phosphatase treatment eliminated the phosphorylated forms, allowing the identification of five distinct bands. Based on their apparent molecular weight, we designated these isoforms from top as the well-characterized long isoform (EEF1D-L), and three short isoforms—EEF1D-M (Medium), EEF1D-N (Normal), and EEF1D-S (Small). The bottom band was non-specific, which was not detected using independent anti-EEF1D antibodies (Fig S1A). Reverse transcription-polymerase chain reaction (RT–PCR) using RNA extracted from mouse liver and skeletal muscle also detected three short splice variants, each producing PCR products of distinct sizes (Fig 1B). Subcloning and sequencing of these PCR products (Fig 1C) confirmed that the variants differ in the inclusion of alternatively spliced exons 5 and 6. Comparative analysis of DNA sequence conservation between the two exons revealed that exon 5 is highly conserved, whereas exon 6 exhibits lower conservation (Fig 1D, top). Furthermore, cross-species alignment of the corresponding amino acid sequences demonstrated that the region encoded by exon 5 is strongly conserved among mammals, suggesting a potentially greater functional importance (Fig 1D, bottom).

Given that the exons 5 and 6 are located between the previously reported LZ and EEF1G-binding domains, alternative splicing may modulate the structure and interaction of these domains, or potentially confer novel interaction capabilities to EEF1D. To identify protein interaction partners of the EEF1D short isoforms, FLAG-tagged variants of EEF1D, -M, -N, and -S were expressed, immunoprecipitated (IP), and analyzed by mass spectrometry (MS) in C2C12 mouse myoblast cells. This IP-MS analysis identified 15, 17, and 13 proteins significantly enriched in EEF1D-N, -M, or -S immunoprecipitates, respectively, compared with the control (Figs 1E and S1B). These enriched proteins included the remaining three components of the EEF1B complex (EEF1B2, EEF1G, and VARS), suggesting that FLAG-tagged EEF1D variants formed the EEF1B complex. Differential interaction analyses among EEF1D variants revealed that a subset of proteins whose binding was dependent on specific exons (Fig 1F and G). Notably, the ER membrane proteins KTN1 and RRBP1, as well as the SNARE protein VAMP7, were significantly reduced in EEF1D-S immunoprecipitates relative to EEF1D-N. Conversely, syntaxin proteins were significantly enriched in EEF1D-M relative to EEF1D-N, suggesting that the domains encoded by exons 5 and 6 are involved in ER-associated functions and vesicle trafficking (20, 21). To extend these findings, we performed an additional IP-MS experiment in mouse hepatoma Hepa1–6 cells expressing FLAG-tagged EEF1D variants. Semi-quantitative analysis confirmed exon 5–dependent interactions with RRBP1 and KTN1 (Fig S1C). In contrast, VAMP7 and syntaxin proteins were detected at low levels across all conditions in Hepa1-6 cells (Table S1).

### The EEF1D–KTN1 interaction is dependent on exon 5

To further validate the exon 5–dependent interaction identified by IP-MS, co-immunoprecipitation followed by immunoblotting was performed. In C2C12 cells, endogenous VAMP7 was not detectably present in either input or co-immunoprecipitated fractions using a commercially available antibody (Fig S1D). In contrast, exon 5–containing variants (EEF1D-M and -N)

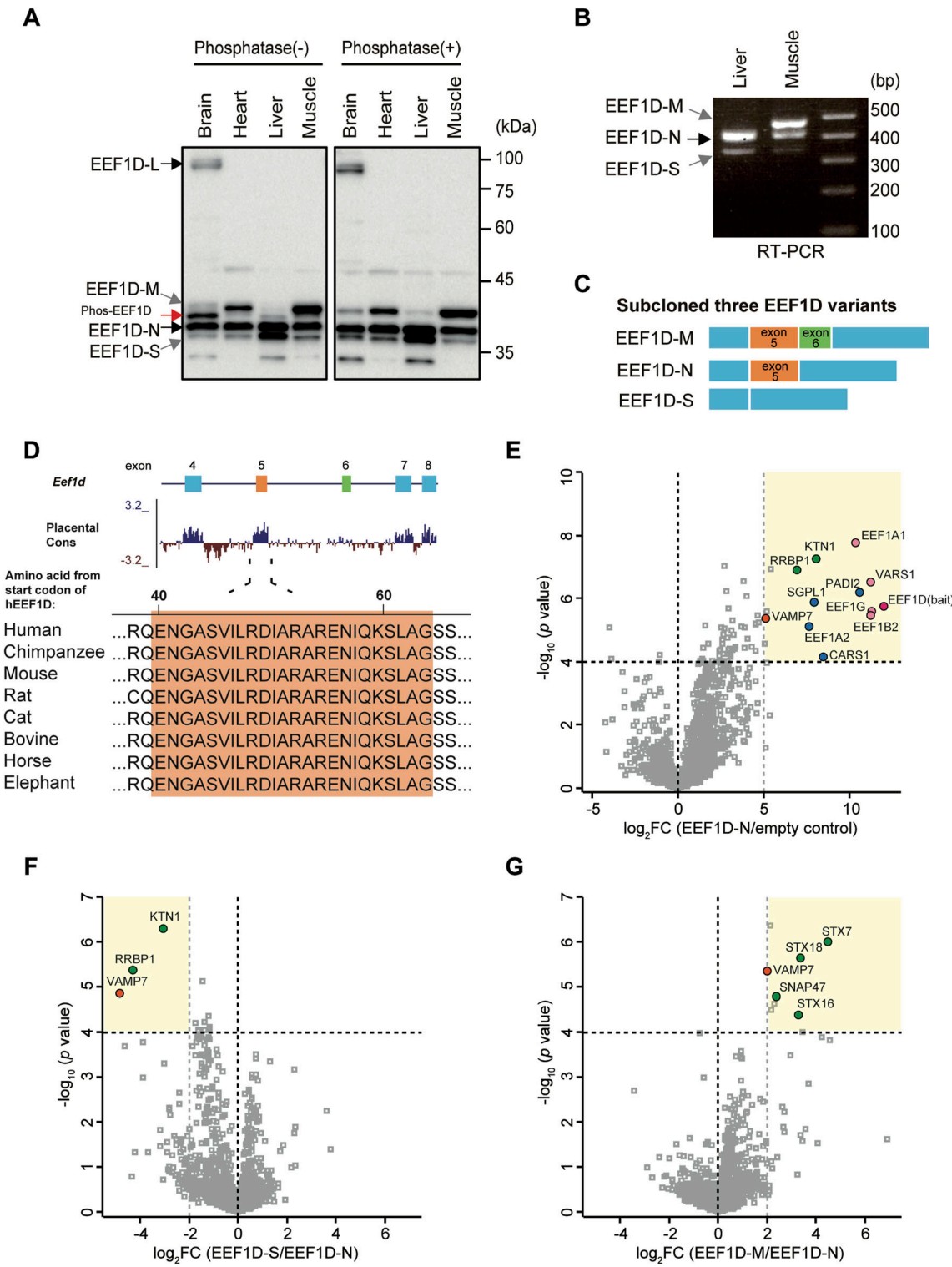

**Figure 1. Identification of EEF1D isoform-specific binding partners.**
**(A)** Immunoblot analysis of EEF1D in various mouse tissues—brain, heart, liver, and skeletal muscle. Four isoforms—EEF1D-L, EEF1D-M, EEF1D-N, and EEF1D-S—are labeled on the left. The EEF1D antibody (10630-1-AP; Proteintech) used in this blot detects both phosphorylated and non-phosphorylated forms. Phosphatase treatment removed phosphorylated forms (phos-EEF1D). Protein amounts loaded for the blot were adjusted to achieve comparable EEF1D signal intensity across tissues. **(B)** Reverse transcription PCR (RT–PCR) analysis of *Eef1d* from mouse liver and skeletal muscle, showing three short *Eef1d* variants. **(C)** Schematic representation of exon usage in the three short *Eef1d* variants based on subcloning of the RT–PCR products. **(D)** A placental mammal conservation (Placental Cons) track, extracted from the UCSC Genome Browser, is shown below the alignment, demonstrating that exon 5 is highly conserved, whereas exon 6 is less conserved (top). Amino acid sequence conservation of EEF1D across multiple animal species. The exon 5-encoded region is highlighted in orange (bottom). **(E)** Comparison of FLAG-IP-MS results between C2C12 cells expressing FLAG-tagged mouse EEF1D-N and empty vector control. The x-axis represents log₂ fold change (FC) of protein enrichment, and the y-axis

specifically co-immunoprecipitated both KTN1 and RRBP1, whereas the exon 5–lacking EEF1D-S isoform did not (Fig S1D and E). Structurally, both KTN1 and RRBP1 are anchored to the ER membrane via a single N-terminal transmembrane domain, with a large coiled-coil domain extending into the cytoplasm (22). Recent studies have revealed that these coiled-coil proteins play critical roles in protein translation (23), shaping peripheral ER morphology (24), and regulating organelle distribution (22), suggesting a potential role for EEF1D in ER-related functions. Among the identified ER-associated interactors, KTN1 exhibited the most reproducible interaction and was, therefore, selected for further functional characterization.

An AlphaFold-predicted structure highlighted the region encoded by exon 5 as a distinct α–helical segment (Fig 2A), suggesting that it may function as a unique structural domain, separate from the well-characterized flanking domains, namely, the upstream EEF1G-binding domain and the downstream LZ domain. To evaluate the contributions of these flanking regions to KTN1 binding and to determine whether this interaction is conserved in human cells, we generated a series of deletion constructs for expression in HEK293T cells. Co-IP assays revealed that deletion of the LZ domain did not disrupt the interaction between EEF1D and KTN1 (Fig 2B), indicating this leucine zipper motif is dispensable for the association. Similarly, removal of the EEF1G-binding domain had no effect on EEF1D–KTN1 binding (Fig 2C). These results demonstrate that the exon 5-encoded region alone is sufficient to mediate interaction with KTN1 and that this mechanism is conserved in human cells.

To confirm this finding under more physiological conditions, we next examined the role of exon 5 at endogenous expression levels. To this end, we established exon 5 KO C2C12 cell lines using CRISPR/Cas9-mediated genome editing (Fig 2D). Guide RNAs flanking exon 5 were used to induce an in-frame deletion, resulting in an EEF1D variant lacking the domain encoded by exon 5. Successful homozygous deletion was confirmed in three independent single-cell clones (Fig 2E). Co-IP assays were performed in WT cells and the validated exon 5 KO clones. Although EEF1D readily co-precipitated KTN1 in WT cells, this interaction was completely abolished in the exon 5 KO clones (Fig 2F). Notably, removal of exon 5 specifically impaired KTN1 binding, as interactions with other known EEF1D partners—including EEF1G, VARS, and EEF1A1—were unaffected in the exon 5 KO cells. Consistent with this, gel filtration analysis revealed a comparable distribution of EEF1B complex components between WT and exon 5 KO cells (Fig 2G), indicating that exon 5 is not required for the overall assembly of the EEF1B complex. Collectively, these findings identify the exon 5-encoded α-helix as a critical and specific determinant for KTN1 recruitment by EEF1D.

### Loss of exon 5 alters the subcellular distribution of the EEF1B complex

Given that the exon 5–encoded domain of EEF1D is essential for its interaction with the ER membrane protein KTN1, we hypothesized that deletion of exon 5 would alter the subcellular localization of

EEF1D and consequently disrupt the localization of the EEF1B complex. To test this, we first examined the overall distribution of EEF1D in WT and exon 5 KO cells. Using WGA to delineate cellular boundaries, we observed a shift in EEF1D distribution from a perinuclear region in WT cells to a more diffuse cytoplasmic distribution in exon 5 KO cells (Fig S2A, top). Pseudocolor imaging further accentuated these changes (Fig S2A, bottom). To directly assess whether this relocalization reflected dissociation from the ER, we performed co-immunostaining of EEF1D with the ER marker SEC61B, a subunit of the SEC61 translocon embedded in the ER membrane (Fig 3A) (25). In WT cells, EEF1D showed pronounced colocalization with SEC61B. In contrast, exon 5 KO cells exhibited a uniform cytoplasmic distribution of EEF1D, lacking the distinct ER-associated pattern observed in WT cells. Quantitative analysis using Pearson's correlation coefficient confirmed a significant reduction in colocalization between EEF1D and SEC61B upon exon 5 deletion (Fig 3B). These results demonstrate that the exon 5 domain is required for robust association of EEF1D with the ER membrane.

To determine whether deletion of exon 5 also affects the localization of the EEF1B complex, we co-immunostained EEF1G with an anti-KDEL antibody, which recognizes the ER retention motifs and serves as an ER marker (26). In WT cells, EEF1D again showed substantial colocalization with KDEL, whereas this overlap was markedly reduced in exon 5 KO clones (Fig S2B and C). Notably, EEF1G was enriched in the ER-associated region in WT cells and exhibited strong colocalization with KDEL (Fig 3C). In contrast, exon 5 KO cells showed a significant reduction in ER-associated enrichment of EEF1G (Fig 3D), indicating the EEF1D–KTN1 interaction is essential for ER localization of the EEF1B complex. To further substantiate the role of the EEF1D–KTN1 interaction in controlling EEF1D localization, we performed rescue experiments using KTN1 KO HeLa cells generated in our previous study (27). Reintroduction of WT KTN1 restored the colocalization of endogenous EEF1D, whereas expression of a KTN1 mutant lacking the EEF1D-binding domain (Δ1D) (10) failed to do so (Fig S2D). These findings demonstrate that the EEF1D-binding domain of KTN1 is indispensable for their colocalization. Taken together, our findings indicate that the association of the EEF1B complex with the ER membrane is largely mediated through the α–helical domain encoded by exon 5 of EEF1D.

### Loss of exon 5 has a minor effect on translation in vitro

Exon 5 KO in C2C12 cells resulted in variable effects on cell growth among independent clones, with a modest but statistically significant reduction observed only in clone 3 (Fig S3A). No measurable defects in nascent protein synthesis were detected, as assessed by puromycin incorporation assays using both immunoblotting and immunostaining (Fig S3B and C). To examine whether deletion of exon 5 affects ER-associated translation, ER-enriched fractions were isolated from C2C12 cells. Although ER fractionation successfully enriched ER-associated proteins, no

represents −log$_{10}$ P-value determined by a t-test. **(F, G)** Differential binding analysis of FLAG-IP-MS data from C2C12 cells expressing FLAG-tagged mouse EEF1D-S relative to EEF1D-N (F), and EEF1D-M relative to EEF1D-N (G), to identify isoform-specific binding proteins.

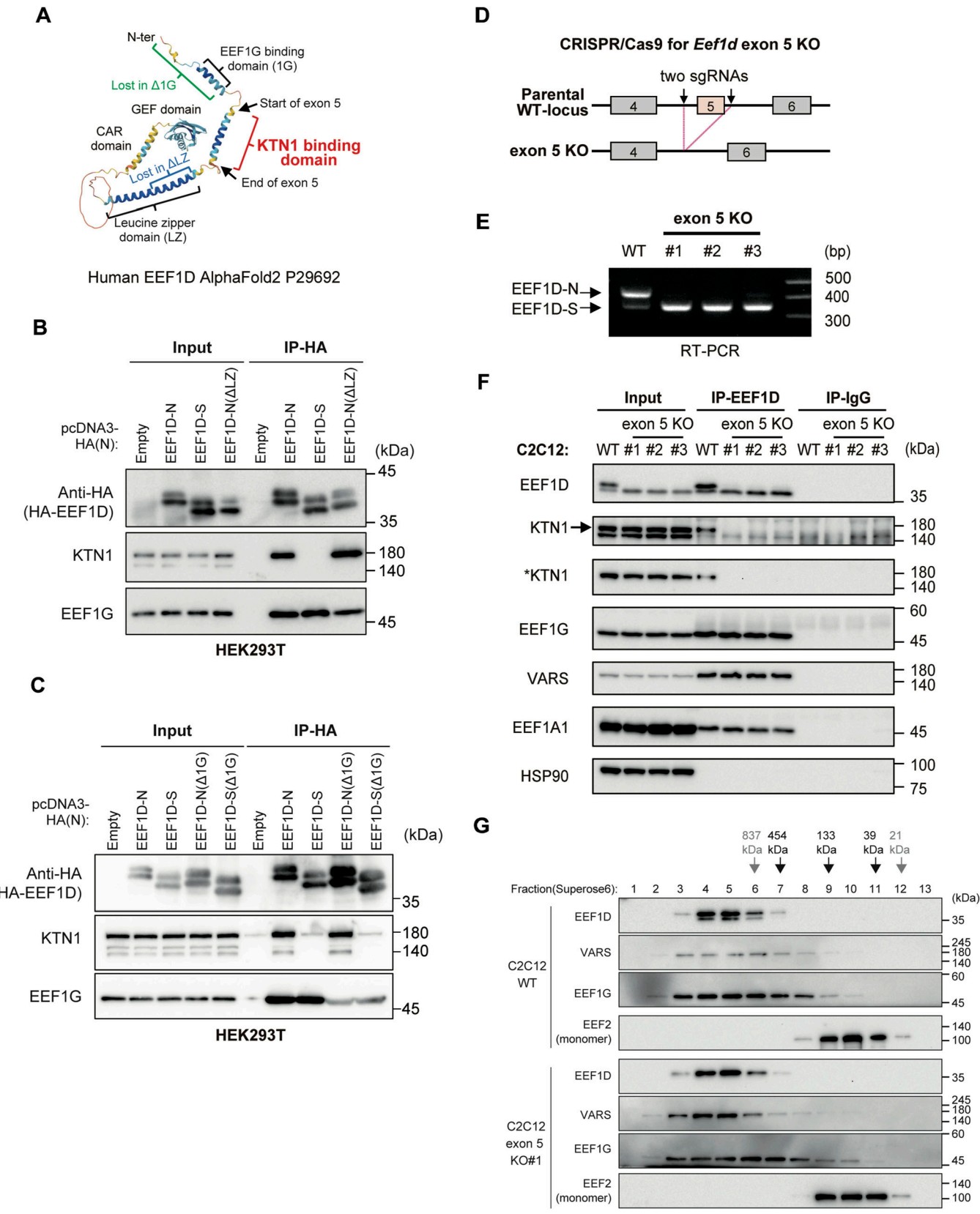

**Figure 2. Exon 5 of EEF1D encodes a binding site of the ER membrane protein KTN1.**
**(A)** Predicted structure of human EEF1D generated by AlphaFold2, showing the region encoded by exon 5 and the known functional domains. **(B)** Immunoblot analysis of HA-IP from HEK293T cells transfected with either empty vector or HA-tagged human EEF1D constructs. ΔLZ denotes the leucine zipper-truncated variant. Input

apparent differences in puromycin incorporation were observed between WT and exon 5 KO C2C12 cells (Fig S3D), indicating that ER-associated translation is not altered by exon 5 deletion. To extend these findings to another cell type, we generated three independent exon 5 KO Hepa1-6 cells. Compared with C2C12 cells, exon 5 KO Hepa1-6 cells showed a slightly greater reduction in cell growth, whereas nascent protein synthesis remained unchanged as assessed by puromycin incorporation assays (Fig 4A and B).

To further evaluate the impact of exon 5 deletion on translation, we performed ribosome profiling (Ribo-seq) (28) using WT and three exon 5 KO Hepa1-6 cells (Fig 4C). Ribo-seq analysis showed the typical hallmarks of ribosome footprints (ribosome-protected fragments, RPFs), including clear three-nucleotide periodicity (Fig 4D) and enrichment within the coding regions (Fig 4E). The distribution of ribosome footprints along coding regions was comparable between WT and exon 5 KO cells (Fig 4E), indicating no major defects in translation elongation. Differential expression analysis revealed that only a small number of genes showed significant changes in translation efficiency (TE), which is defined as ribosome footprint counts normalized by RNA-seq counts (Fig 4F). Moreover, translation of ER-targeted transcripts harboring signal peptides was not substantially altered in exon 5 KO cells (Fig 4G). Collectively, these results suggest that exon 5 is largely dispensable for translational regulation in both the global and ER-associated compartments under in vitro conditions.

### Loss of exon 5 alters the subcellular distribution of EEF1D in mouse tissues

To investigate the in vivo role of exon 5 in EEF1D, we generated an exon 5 KO mouse model using the CRISPR/Cas9 system. Two independent exon 5 KO lines were established, and Sanger sequencing confirmed precise removal of the targeted exon in both lines (line #1 and #2) (Fig S4A–C). RT–PCR analysis of multiple tissues revealed that homozygous exon 5 KO mice specifically lacked the larger transcripts and retained only the shorter transcripts corresponding to the engineered deletion (Fig 5A). Deletion of exon 5 was further confirmed at the protein level (Fig S4D). These results confirm the successful generation of two independent mouse lines carrying in-frame deletions of exon 5. All subsequent analyses were performed using the homozygous exon 5 KO mouse line #1 unless otherwise specified.

Data from the International Mouse Phenotyping Consortium (IMPC) indicate that homozygous KO of *Eef1d* is embryonically lethal, whereas *Eef1b2* KO is not, highlighting the essential role of EEF1D in embryonic survival (Fig 5B, top). In contrast, EEF1D exon 5 KO mice were viable (Fig 5B, bottom) and exhibited no substantial differences in body weight from 3 to 9 wk of age compared with WT controls (Fig 5C). Moreover, hematological analyses showed no

significant differences in white blood cell (WBC) count, RBC count, hemoglobin (Hgb), hematocrit (Hct), or platelet parameters between exon 5 KO and WT mice (Fig S4E), indicating that the exon 5-specific functions are dispensable for survival and hematopoiesis under basal conditions.

To explore the cellular and tissue-level consequences of exon 5 deletion, we next examined its impact in primary MEFs (Fig S5A). Co-IP experiments using lysates from WT and exon 5 KO MEFs revealed that the interaction between EEF1D and KTN1 was abolished in exon 5 KO cells, whereas interactions with other EEF1B complex components, including EEF1G and VARS, were preserved (Fig S5B). Consistent with our observations in C2C12 cells, immunostaining analysis showed that EEF1D co-localized with the ER marker KDEL in WT MEFs, whereas in exon 5 KO MEFs, EEF1D displayed a more diffuse cytoplasmic distribution (Fig S5C).

Next, to determine whether exon 5 regulates EEF1D localization in vivo, we examined the small intestine and liver as representative tissues using immunohistochemical and immunofluorescence analyses. The small intestinal epithelium is highly polarized, with clearly defined apical, basal, and ER-associated regions, allowing precise assessment of protein distribution relative to the ER. The liver was selected because of its high protein synthesis activity. In the small intestine, SEC61B—an ER marker—exhibited predominant localization in the apical region in both WT and exon 5 KO mice (Fig 5D, top), indicating that the overall ER distribution was not altered by exon 5 deletion. Consistent with this, electron microscopy analysis of small intestinal epithelial cells revealed no obvious differences in ER distribution or morphology between WT and exon 5 KO mice (Fig S6A), suggesting that loss of EEF1D exon 5 does not disrupt ER organization. In WT mice, small intestinal epithelial cells displayed strong EEF1D staining with a clear apical and ER-associated pattern, whereas epithelial cells from exon 5 KO mice showed diffuse cytoplasmic localization of EEF1D (Fig 5D bottom and Fig S6B). Consistent with our mouse data, immunohistochemical data from The human protein atlas showed that human small intestine tissue exhibits similar staining patterns for EEF1D, KTN1, and SEC61B (Fig S6C), providing additional support for ER localization of EEF1D in small intestinal epithelial cells. Similarly, hepatocytes from exon 5 KO mice exhibited loss of the localized EEF1D staining observed in WT hepatocytes (Figs 5E and S6D), whereas SEC61B distribution remained comparable between WT and exon5 KO tissues. Together, these results indicate that exon 5 enhances ER-associated localization of EEF1D in mouse tissues.

### Deletion of exon 5 in mouse tissues shows tissue-specific down-regulation of the EEF1B complex

Given that exon 5 deletion altered EEF1D localization in the liver, we next sought to identify potential molecular alterations using

represents total cell lysates before IP. **(C)** Immunoblot analysis of HA-IP from HEK293T cells transfected with either empty vector or HA-tagged EEF1D constructs. Δ1G denotes deletion of the EEF1G-binding region. **(D)** Schematic representation of *Eef1d* exon 5 KO cell generation using the CRISPR/Cas9 system. Squares represent exons, and black arrows indicate Cas9 cleavage sites within intron 4 and intron 5. **(E)** RT–PCR analysis of *Eef1d* using cDNA from WT and exon 5 KO C2C12 cells. **(F)** Immunoblot analysis of EEF1D-IP and control IgG-IP from WT and exon 5 KO C2C12 cells. Loss of EEF1D–KTN1 interaction was assessed using two KTN1 antibodies: KTN1 (19841-1-AP; Proteintech) and *KTN1 (sc-136534; Santa Cruz Biotechnology). The EEF1D antibody (SC-393731; Santa Cruz Biotechnology) used for detection does not recognize phosphorylated EEF1D. **(G)** Immunoblot of fractions separated by gel filtration on Superose 6 using WT cells and exon 5 KO #1 clone C2C12 cells. Estimated molecular sizes (top) were determined using standard proteins. Black values fall within the calibrated range, whereas gray values are outside the range.

**A**

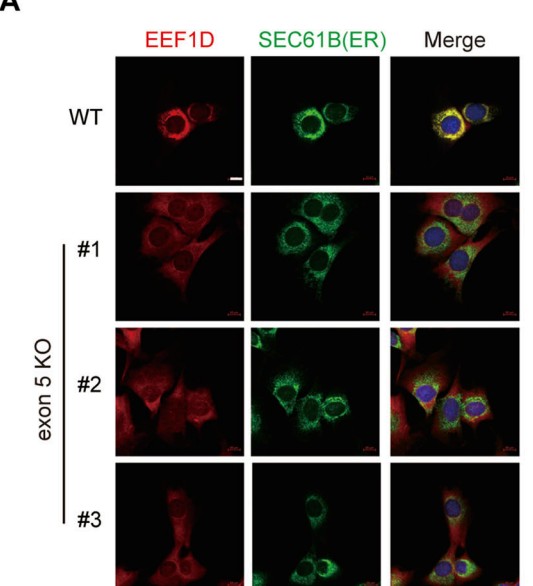

**B**

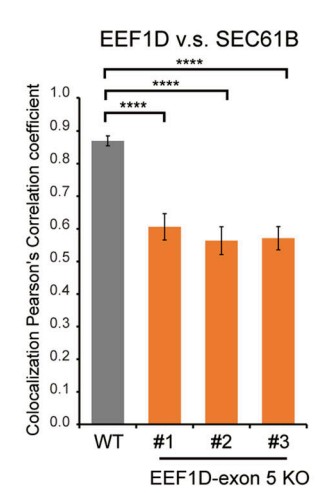

**Figure 3. EEF1D localizes to the ER in an exon 5-dependent manner.**
**(A)** Representative confocal immunofluorescence images of WT and exon 5 KO C2C12 cells stained with anti-EEF1D (SC-393731; Santa Cruz, red), anti-SEC61B (green), and DAPI (blue). Scale bar, 10 $\mu$m. **(B)** Pearson's correlation coefficient between EEF1D and SEC61B signals in WT and exon 5 KO C2C12 cells. Bars represent mean ± SD (n = 21) with $P$-values determined by one-way ANOVA, Tukey's post hoc test (****$P$ < 0.0001). **(C)** Same as (A), but cells were stained with anti-EEF1G (red) and anti-KDEL (green). Scale bar, 10 $\mu$m. **(D)** Same as (B), but showing Pearson's correlation coefficient between anti-EEF1G and anti-KDEL signals (n = 23).

**C**

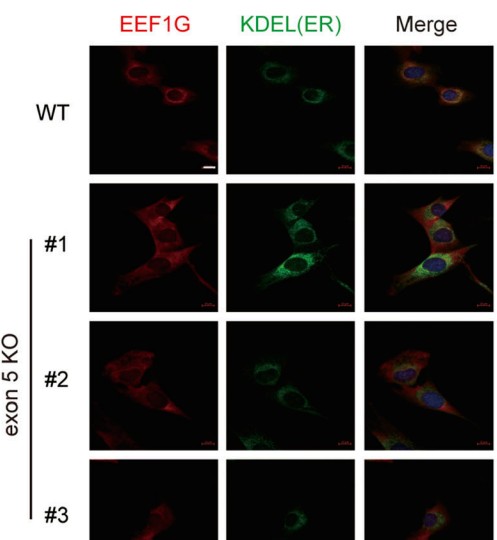

**D**

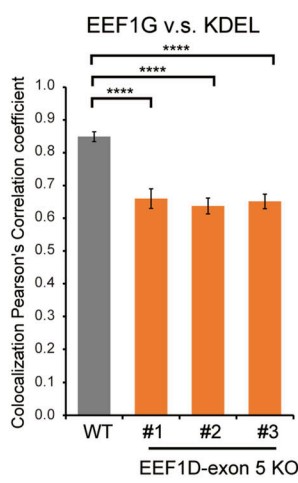

whole-proteome analysis of liver tissue. In addition to liver samples, which served as a representative in vivo model, MEFs were analyzed to provide a complementary in vitro perspective. Whole-proteome analysis of liver tissue revealed significant downregulation of EEF1B complex components, including EEF1D, as well as the anchor protein KTN1, in exon 5 KO samples compared with WT controls (Fig 6A, left). The protein level of EEF2, another key translation elongation factor, was unchanged, confirming the specificity of this effect for the EEF1D-containing complex. Notably, whole-proteome analysis of MEFs showed no significant differences between WT and exon 5 KO cells, suggesting that

this phenotype occurs in a context-dependent manner (Fig 6A, right).

To determine whether this effect extends beyond the liver, we analyzed heart, skeletal muscle, brain, and small intestine tissues by immunoblotting for components of the EEF1B complex. Immunoblot analysis of an independent cohort of mice confirmed reduced levels of EEF1B complex proteins in the livers of exon 5 KO mice (Fig 6B). Decreased levels of EEF1D, VARS, and EEF1G were also observed in the heart and skeletal muscle (Fig 6C, left two panels). In contrast, no reductions in EEF1B complex components were detected in the brain or small intestine (Fig 6C, right two panels).

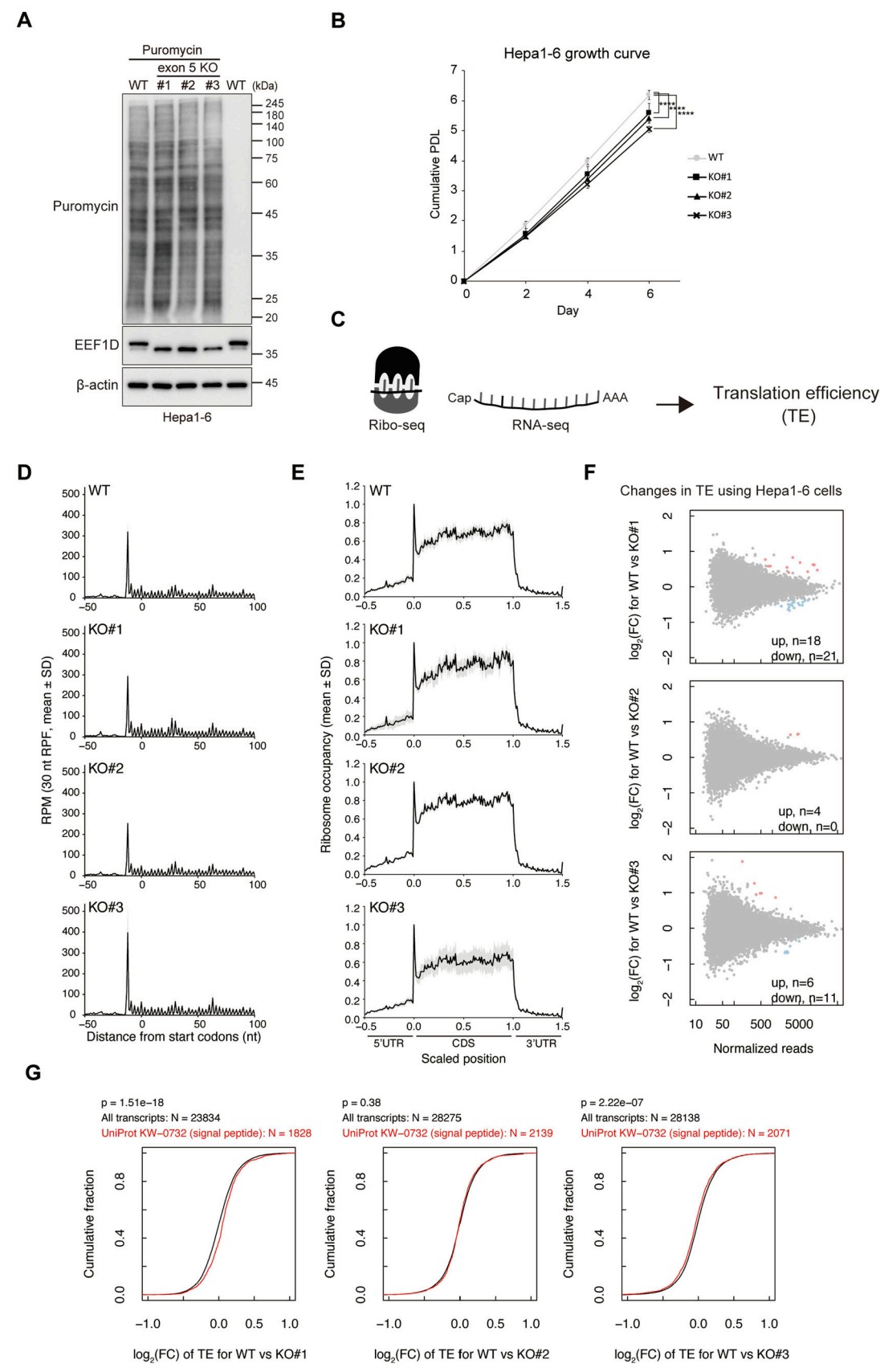

Notably, although protein levels of EEF1B complex components were reduced in exon 5 KO livers, mRNA levels of *Eef1d*, *Eef1g*, *Eef1b2*, and *Vars* remained unchanged (Fig 6D), indicating that this reduction occurs at the post-transcriptional level. These results demonstrate that down-regulation of the EEF1B complex in exon 5 KO mice occurs in a tissue-specific manner.

We next assessed the functional impact of EEF1B complex down-regulation on hepatic protein synthesis using a puromycin in-corporation assay. Despite reduced protein levels of the EEF1B complex, no significant differences in global translation were observed between WT and exon 5 KO livers (Fig S7A and B). This finding suggests that the residual EEF1B complex is sufficient to sustain protein synthesis under the experimental conditions ex-amined, or alternatively, that compensatory mechanisms act to preserve translational activity. In addition, we conducted plasma biochemical analyses to assess key indicators of hepatic synthetic function and metabolism status, including albumin, total protein, glucose, and triglyceride levels. Consistent with the translational analyses, no significant differences were observed between exon 5 KO and WT mice (Fig S7C), indicating that deletion of exon 5 does not impair liver function under physiological conditions.

Taken together, although the phenotypic consequences of exon 5 KO are subtle, our findings support a mechanistic model in which alternative splicing of EEF1D governs the subcellular positioning of the EEF1B complex to the ER through interaction with KTN1. Con-versely, exclusion of exon 5 abolishes this interaction, resulting in relocalization of the EEF1B complex from the ER to the cytoplasm, which is associated with a tissue-specific reduction in complex abundance (Fig 6E).

## Discussion

Our data reveal that exclusion of exon 5 from *Eef1d* through al-ternative splicing disrupts anchoring of the EEF1B complex to the ER membrane via KTN1. This finding uncovers a previously un-recognized layer of translational regulation in animals, whereby alternative splicing of a core elongation factor controls its mo-lecular interactions, protein abundance, and subcellular locali-zation. Notably, the expression ratio of EEF1D splice variants—with or without exon 5—varies among mouse tissues, suggesting that the extent of EEF1D association with the ER may contribute to tissue-specific functions. Furthermore, the exon 5 KO mouse model generated in this study provides a valuable tool for investigating the physiological consequences of disrupting this molecular an-choring mechanism in vivo.

Previous work by Ong et al demonstrated that residues 1,125–1,165 of KTN1 interact with EEF1D; however, the specific region of EEF1D responsible for this interaction remained unidentified (10). Our findings reveal that the α-helix encoded by exon 5 con-stitutes the KTN1-binding interface. This conclusion is supported by a concurrent study by Teixeira et al (29), which used structural prediction and biochemical assays using recombinant proteins to identify the N-terminal helix of EEF1D (residues 30–66, corre-sponding to the exon 5-encoded helix) as the KTN1 binding site. Although our study primarily focuses on the interaction between EEF1D and KTN1, we also identified differential binding of another ER-associated protein, RRBP1, suggesting that RRBP1, together with KTN1, may regulate the association of the EEF1B complex and the ER. Furthermore, our data demonstrate that deletion of exon 5 abolishes ER anchoring of the entire EEF1B complex. Relocali-zation of the EEF1B complex in exon 5 KO cells was consistently observed across multiple systems, including immortalized cell lines, primary MEFs, and mouse tissues. Human intestinal epi-thelial cells also exhibited colocalization of EEF1D, KTN1, and ER markers, supporting conservation of this mechanism across species. Collectively, our results identify exon 5 as a previously unrecognized KTN1-binding module that governs subcellular po-sitioning of the EEF1B complex.

Given that intracellular mRNA localization and localized translation are widely used to achieve spatial control of protein synthesis (30), changes in EEF1D localization may influence translation in a spatially restricted manner. Indeed, disruption of the KTN1–EEF1D interaction has been shown to reduce membrane protein expression, supporting the idea that KTN1 anchors translational regulators to the ER to locally promote protein synthesis (23). However, in our analyses, despite clear relocali-zation of EEF1D in exon 5 KO cells and tissues, puromycin-labeling assays revealed no significant differences in global translation rates compared with WT controls. These findings suggest that ER association of the EEF1B complex does not play a dominant role in regulating overall protein synthesis under basal conditions. Furthermore, we did not detect altered translation efficiency of transcripts encoding signal peptides in our Ribo-seq analysis using Hepa1-6 cells, nor did we observe changes in serum al-bumin levels in exon 5 KO mice. Consistent with these obser-vations, we recently reported that KTN1 recruits aminoacyl-tRNA synthetases to the ER membrane and that disruption of this

**Figure 4. Loss of exon 5 has limited effects on translation in vitro.**
**(A)** Representative immunoblot of a puromycin incorporation assay in WT and exon 5 KO Hepa1-6 cells. The rightmost lane is a puromycin-negative control. β-actin served as a loading control. **(B)** Growth curves of WT and exon 5 KO Hepa1-6 cells. Cell numbers were measured every 2 d. Data represent the mean with SD of three replicates. ****$P < 0.0001$, determined by two-way ANOVA with Sidak's test. **(C)** Schematic representation of Ribo-seq and RNA-seq experiments to assess the effect of exon 5 deletion on translation in Hepa1-6 cells. **(D)** Metagene analysis of Ribo-seq reads around the start codon. The x-axis represents the nucleotide position of the 5' end of 30-nt length ribosome-protected fragments (RPFs) relative to the start codon, and the y-axis represents reads per million of RPFs. The mean and ±SD across replicates are shown as a black line and a gray shaded area, respectively. **(E)** Metagene analysis of ribosome occupancy normalized to ribosome accumulation at the start codon. RPFs were first adjusted to the A-site position (A-site offset), and transcripts with at least one RPF per codon were subsequently included in the metagene analysis. The x-axis represents the relative position of the scaled 5' UTR, CDS, and 3' UTR regions (the lengths of UTRs and CDS are set to 0.5 and 1, respectively). The mean and ±SD across replicates are shown as a black line and a gray shaded area, respectively. **(F)** MA plots showing the $\log_2$ fold-change (FC) of translation efficiency (TE) for each transcript in exon 5 clone cells relative to WT cells, together with normalized read counts. The up- or down-regulated transcripts (adj $P$-value < 0.05) are highlighted in pink or blue, respectively, and their numbers are indicated. **(G)** Cumulative fraction curves showing the $\log_2$ FC of TE in exon 5 KO clone cells relative to WT cells for all transcripts (black) and a subset of transcripts harboring signal peptides (red). Statistical significance was assessed using a two-sided Wilcoxon rank-sum test.

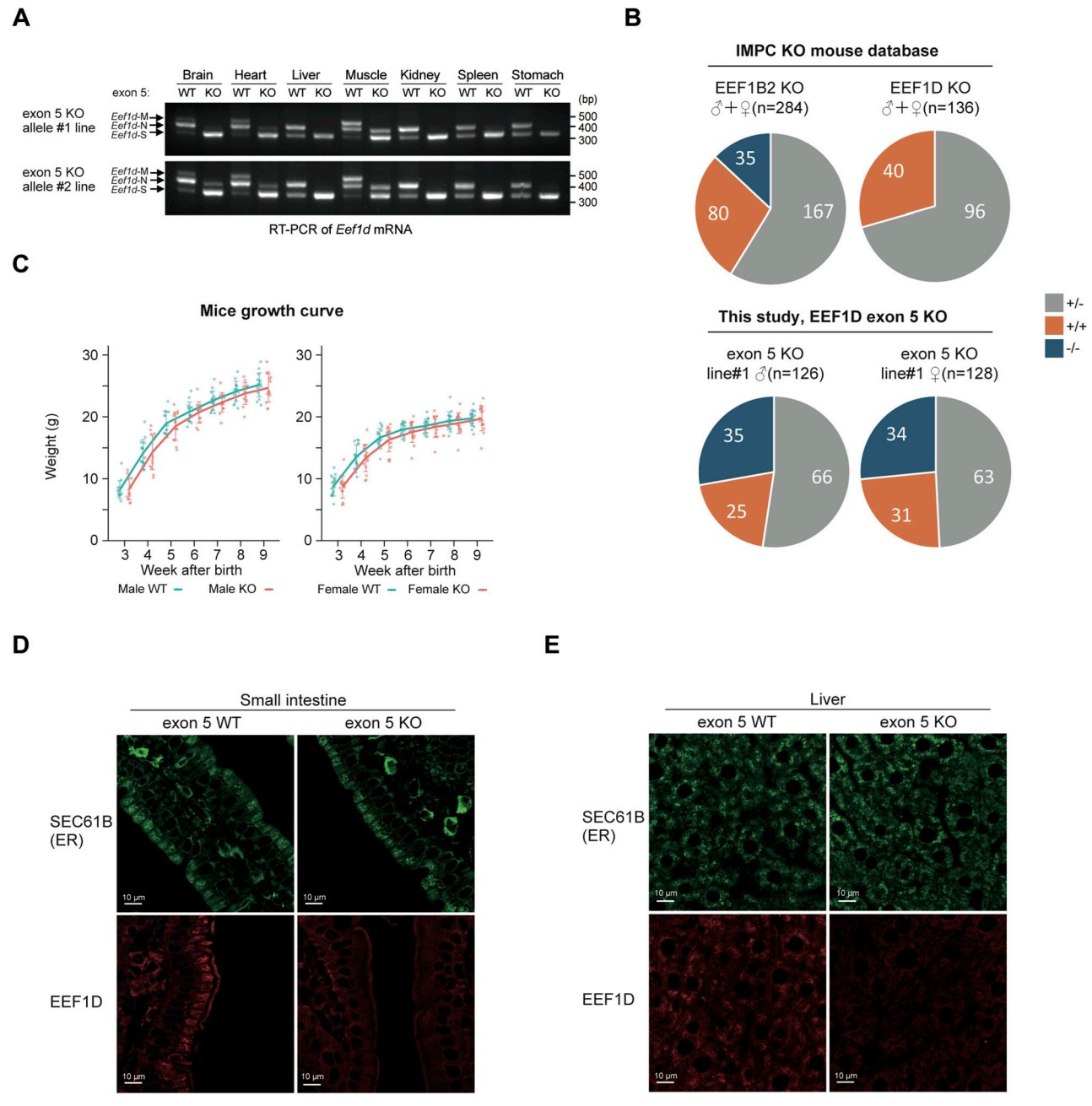

**Figure 5. Relocalization of EEF1D in exon 5 KO mice.**
**(A)** RT–PCR analysis of *Eef1d* mRNA expression in various tissues from WT, homozygous KO line #1, and homozygous KO line #2 mice. The gel shows the PCR products corresponding to three short EEF1D variants. **(B)** Pie charts showing the birth rate distribution of WT, heterozygous, and homozygous KO genotypes. The top row shows data from the IMPC KO mouse database for *Eef1b2* KO and *Eef1d* KO mouse lines. The bottom row shows data from this study for the *Eef1d* exon 5 KO line #1, with male mice (left) and female mice (right) presented separately. The total number of mice (n) is indicated for each chart. **(C)** Growth curves for male (left) and female (right) WT and exon 5 KO mice. Body weight is plotted against age in weeks after birth. Mean ± SD and individual data points are shown for each group. **(D, E)** Confocal immunofluorescence images of mouse small intestine (D) and liver (E) showing the localization of SEC61B (green, top panels) and EEF1D (10630-1-AP; Proteintech, red, bottom panels) in WT and exon 5 KO mice. Scale bar, 10 μm.

localization has only a limited impact on global translation (27). Together, these findings suggest that ER localization of translation regulators, including the EEF1B complex, contributes to protein synthesis in a context-dependent manner rather than previously appreciated.

The abundance of protein complex components is often co-ordinated at the post-translational level to maintain stoichiometry (31, 32). Our proteomic and immunoblot analyses revealed that liver, heart, and skeletal muscle tissues from exon 5 KO mice exhibited uniformly reduced protein levels of EEF1D, together with

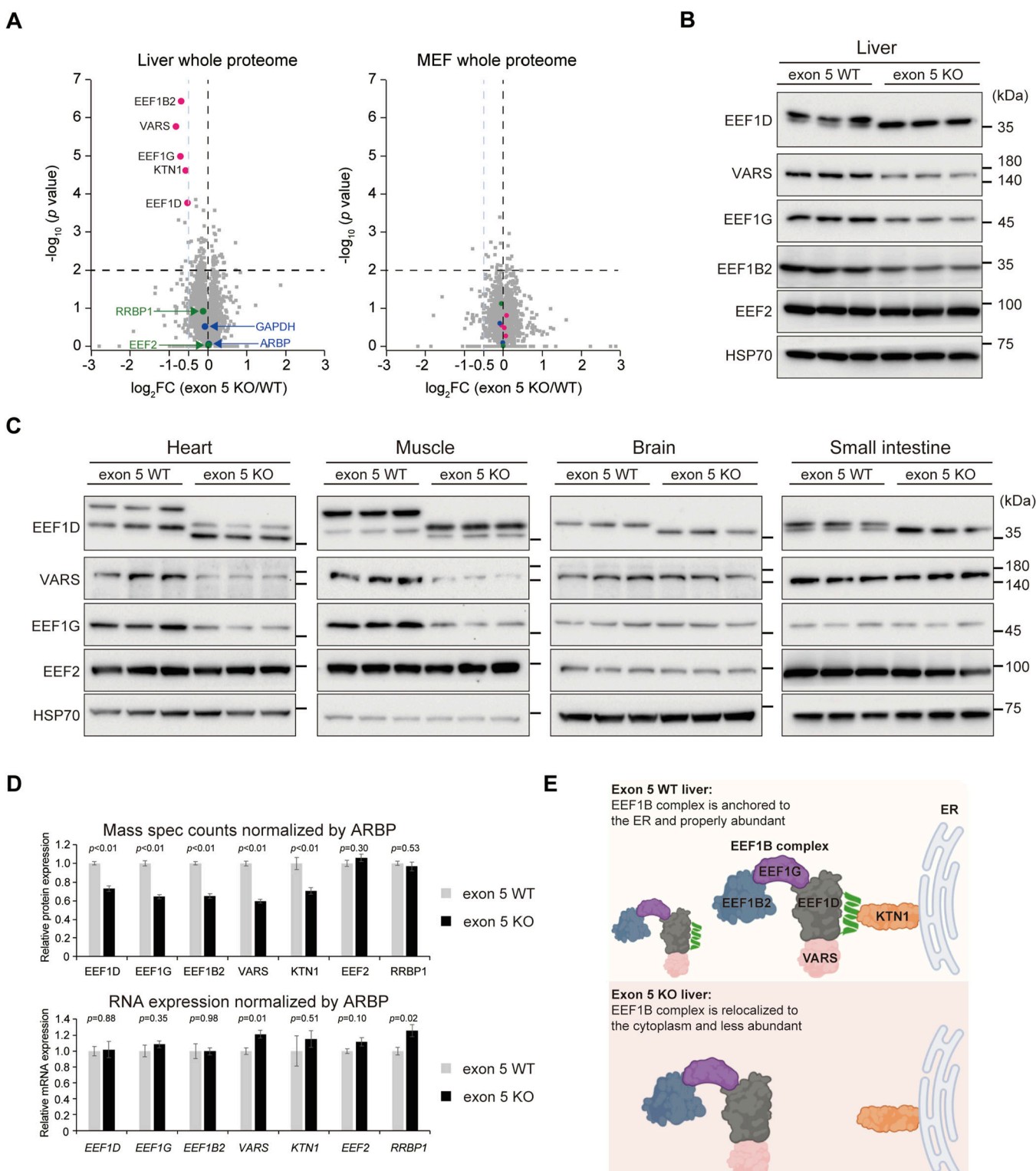

**Figure 6. Loss of exon 5 leads to a decrease in the EEF1B complex in a tissue-specific manner.**
**(A)** Volcano plots of whole proteome analysis from WT and exon 5 KO liver (left) and MEFs (right). The x-axis represents the $\log_2$ FC in protein abundance between exon 5 KO and WT samples, and the y-axis represents the statistical significance as $-\log_{10}$ ($P$-value). The horizontal dashed line indicates the significance threshold ($P = 0.01$). Highlighted proteins include components of the EEF1B complex and KTN1 (magenta) and control proteins (green, blue). **(B)** Immunoblot analysis of EEF1B complex components in liver lysates from WT and exon 5 KO mice. EEF2 and HSP70 served as internal controls. EEF1D antibody (SC-393731; Santa Cruz Biotechnology) was used for detection. **(C)** Immunoblot analysis as in (B), using heart, skeletal muscle, brain, and small intestine lysates from WT and exon 5 KO mice. **(D)** Bar graphs showing relative protein expression from mass spectrometry counts (top) and relative mRNA expression determined by quantitative RT–PCR (bottom) for selected genes in WT and exon 5 KO liver, normalized to ARBP. Bars represent mean ± SEM, and $P$-values determined by $t$-test for genotypes comparison are shown above each pair (n = 5).

its canonical complex partners (EEF1G, EEF1B2, VARS) and KTN1. Interestingly, this down-regulation was not observed in MEFs, brain, and small intestine tissues, indicating that stability of the dissociated EEF1B complex may be differentially regulated or compensated in a tissue-specific manner. Although the EEF1B complex was mislocalized and reduced in abundance in multiple tissues of exon 5 KO mice, these molecular alterations did not result in measurable defects in early development or growth in young animals. Notably, genetic disruption of the large isoform, EEF1D-L, in mice has been reported to result in normal behavior in basal conditions, but to impair locomotor activity after fear conditioning (15). Furthermore, protein synthesis is known to be sensitive to cellular stress, and reduced availability of aminoacyl-tRNAs under stressed conditions can compromise translation elongation (33, 34). It, therefore, remains possible that ER tethering of the EEF1B complex becomes functionally important under specific physiological or stress conditions. Future studies involving ER stress, metabolic challenge, or aging may help clarify the context-dependent relevance of exon 5-mediated regulation of EEF1D localization and abundance.

Because the ER is a multifunctional organelle beyond its role in protein translation, recruitment of EEF1D to the ER may influence additional ER-associated processes. KTN1 has been reported to act as a critical scaffold that organizes diverse binding partners involved in ER dynamics (22). These partners include the plus-end directed motor kinesin-1, the dynein adaptor protein CDR2, and the small GTPase RhoG, linking ER positioning to the cytoskeletal network (29, 35, 36). Recent studies have shown that CDR2 and its homolog CDR2L regulate ER morphology via KTN1 binding, and that this interaction competes with EEF1D binding to KTN1 (29). Although these findings raise the possibility that exon 5 KO mice may exhibit alterations in ER structure or organization, our electron microscopy analysis did not reveal overt ER structural abnormalities in the small intestine. In addition to these KTN1-associated pathways, our IP-MS identified the SNARE protein VAMP7 as an exon 5-dependent binding partner of EEF1D, as well as multiple SNARE proteins as exon 6-associated proteins. VAMP7 and SNARE proteins play essential roles in vesicle transport along microtubules, endosomal sorting, and autophagy (37, 38). Because vesicle trafficking is closely linked to ER function and membrane dynamics, these findings suggest a potential connection between alternative splicing of EEF1D and broader membrane-associated processes. Elucidating these aspects will enhance our understanding of how tissue-specific alternative splicing of EEF1D coordinates the translational machinery with membrane-associated dynamics.

A limitation of this study is that we did not determine the absolute ratio of soluble to ER membrane-associated EEF1D or the molar ratio of KTN1 to EEF1D. Semi-quantitative densitometric analysis of EEF1D band intensity in Fig S1A suggested that the relative ratios of EEF1D-S to EEF1D-N in liver were ~1:0.4–0.7 across three independent antibodies. However, these data do not distinguish the fraction of EEF1D-N associated with the ER membrane,

and therefore do not provide a direct measure of ER membrane-bound versus soluble pools. In addition, accurate estimation of the molar ratio between KTN1 and EEF1D would require cross-comparison between different antibodies, which is technically challenging and was not addressed in this study. These parameters are likely to vary depending on cell type, tissue context, and physiological state and will require dedicated quantitative biochemical or proteomic analyses in future studies.

## Materials and Methods

### Cell culture

C2C12 myoblast cells were obtained from American Type Culture Collection (CRL-1772; ATCC), cultured in DMEM (Cat# 08459-64; Nacalai Tesque) with 20% FBS (Gibco) and 50 U/ml Penicillin and Streptomycin (Nacalai Tesque) at 37°C in a humidified 5% $CO_2$ incubator. Hepa1–6 cells were obtained from RIKEN BRC (RCB1638), cultured in DMEM with 10% FBS and 50 U/ml Penicillin and Streptomycin, 1% MEM-nonessential amino acids (100×, Nacalai Tesque), 2 mM L-glutamine (Nacalai Tesque), 1 mM sodium pyruvate (Nacalai Tesque) at 37°C in a humidified 5% $CO_2$ incubator. HEK293T cells (ATCC; CRL-3216) and HeLa cells (ATCC; CCL-2) were cultured in the same conditions as Hepa1-6 cells. The cell growth curve was measured by cell counting every 2 d. The degree of cell proliferation of C2C12 and Hepa1-6 cells was represented as the accumulated population doubling level ($log_2$ [cell number after 2 d/initial seeded cell number]).

### Subcloning, plasmids, and expression

The cDNA encoding the short isoforms of mouse EEF1D was amplified by PCR from samples of postnatal day 28 mouse (C57BL/6, male) liver and skeletal muscle. The coding sequence (CDS) of the short isoforms—designated N (281 amino acid, Consensus CDS [CCDS] ID: CCDS27554.1), S (257 amino acid, CCDS70638.1), and M (300 aa, CCDS ID not registered, but corresponding to RefSeq IDs: NP_001398895, NP_001398896, NP_001398897)—were verified by Sanger sequencing and subsequently subcloned into the lentiviral expression vector pCSIIpuro with an N-terminal FLAG-tag. The resulting constructs and pCSIIpuro-empty control were introduced into C2C12 and Hepa1–6 via lentivirus infection. Lentivirus was produced by co-transfecting HEK293T cells with each expression construct, the packaging plasmid pPAX2 (#12260; Addgene), and the envelope plasmid pMD2.G (#12259; Addgene) using PEI-MAX polyethyleneimine transfection reagent (Polysciences). The lentiviral supernatant was collected 48 h post-transfection, supplemented with 4 µg/ml polybrene (Sigma-Aldrich), and applied to C2C12 and Hepa1-6 cells for 24 h. Infected cells were then selected with puromycin (Sigma-Aldrich) to establish stable cell lines expressing the respective constructs.

---

(E) Model illustrating the role of EEF1D exon 5 in mouse liver. In the exon 5 WT state (top), the EEF1B complex is anchored to the ER via interaction between EEF1D and KTN1. In the exon 5 KO state (bottom), loss of this interaction causes relocalization of the complex from the ER to the cytoplasm and reduces its stability.

The CDS of human EEF1D with exon 5 inclusion (EEF1D-N), exon 5 exclusion (EEF1D-S), and partial leucine zipper deletion (EEF1D-N[ΔLZ]) were obtained from plasmids FXC34505E, FXC34503E, and FXC34504E, respectively, from the Kazusa cDNA clone collection (Kazusa DNA Research Institute, https://www.kazusa.or.jp/). Deletion of the EEF1G binding domain (Δ1G) was achieved by truncating aa 1–28 from the CDS of EEF1D-N and EEF1D-S, resulting in the constructs EEF1D-N(Δ1G) and EEF1D-S(Δ1G). All CDS variants were subcloned into the mammalian expression vector pcDNA3 with an N-terminal HA-tag. These constructs were transiently transfected into HEK293T cells using PEI-MAX reagent.

The CDS of human KTN1 WT lacking a stop codon was obtained from plasmid FXC11648 from the Kazusa cDNA clone collection. To delete the EEF1D binding domain (Δ1D) from KTN1 (KTN1 Δ1D), the EEF1D binding domain (ADEMHTLLQLECEKYKSVLAETE-GILQKLQRSVEQEENKWK), as previously reported (10), was deleted using the In-Fusion HD Cloning Kit (Takara). Both CDS constructs were then transferred to the doxycycline-inducible PiggyBac plasmid PB-TA-ERN-FLAG(C) (a modified version of Addgene plasmid #80474, with a C-terminal FLAG sequence inserted), resulting in expression plasmids encoding C-terminal FLAG-tagged proteins. To generate HeLa cells expressing either KTN1 WT or KTN1 Δ1D, previously generated KTN1-KO HeLa cells (27) were rescued via co-transfection of the respective PB-TA-ERN-FLAG(C)-KTN1 constructs along with the Super PiggyBac Transposase Expression Vector (System Biosciences) using FuGENE HD (Promega). After transfection, cells were selected with 600 µg/ml G418 for 1 wk, and then treated with 1 µg/ml doxycycline (D5897; LKT Laboratories) for 3 d to induce expression.

### Generation of EEF1D exon 5 KO cell line by the CRISPR-Cas9 system

Single guide RNAs (sgRNAs) were designed using the CRISPR Targets Track from UCSC Genome Browser (https://genome.ucsc.edu/) to target intron 4 and intron 5 of the *Eef1d* genomic sequence. Two different sgRNAs targeting intron 4 (designated S1 and S2) and two targeting intron 5 (S3 and S4) were individually subcloned into the vector pSpCas9(BB)-2A-puro (#48139; Addgene), resulting in the constructs pSpCas9-S1 (pS1), pSpCas9-S2 (pS2), pSpCas9-S3 (pS3), and pSpCas9-S4 (pS4). The oligonucleotide sequences used for sgRNA cloning (S1-F/R, S2-F/R, S3-F/R, and S4-F/R) are listed in Table 2. To delete exon 5, the following plasmid combinations: pS1+pS3, pS1+pS4, pS2+pS3, and pS2+pS4, were transfected into C2C12 and Hepa1-6 cells using Lipofectamine 3000 (Thermo Fisher Scientific). 2 d after transfection, cells were selected with puromycin for 3 d. The resulting heterogeneous population was then subjected to single-cell cloning via limiting dilution cloning. Individual clones were screened for successful homozygous exon 5 deletion by PCR and Sanger sequencing using the following primers: 5′-GATTCTAT-GAGCAGATGAACGG-3′ and 5′-CTTCCAAGGCATGTGGCAAC-3′. The plasmid combinations used for each clone were as follows. For C2C12 exon 5 KO cells: clone #1 (pS1 + pS3), clone #2 (pS1 + pS4),

**Table 2. Oligonucleotides used for sgRNA targeting exon 5.**

| Target | Sequence (5′-3′) |
|---|---|
| S1-F | CACCGGCAAGCAACAGGTTGTCTCA |
| S1-R | AAACTGAGACAACCTGTTGCTTGCC |
| S2-F | CACCGGGTGGGCTAGAGACATGTTA |
| S2-R | AAACTAACATGTCTCTAGCCCACCC |
| S3-F | CACCGTGTGCCCTAACTTGCTCCAA |
| S3-R | AAACTTGGAGCAAGTTAGGGCACAC |
| S4-F | CACCGAGTTCGTCAGTGCCTGCCGG |
| S4-R | AAACCCGGCAGGCACTGACGAACTC |

and clone #3 (pS2 + pS4). For Hepa1-6 exon 5 KO cells: clone #1 (pS1 + pS4), clone #2 (pS1 + pS4), and clone #3 (pS2 + pS4).

### Generation of EEF1D exon 5-deficient mice

EEF1D exon 5-deficient mice were generated by improved genome editing via oviductal nucleic acid delivery (i-GONAD) method, as previously described (39). Briefly, based on our prior evaluation of EEF1D exon 5 KO efficiency in cell lines, we designed the following two CRISPR RNAs (crRNAs) flanking exon 5: 5′-GCAAGCAAC AGGUUGUCUCA-3′ and 5′-UGUGCCCUAACUUGCUCCAA-3′. The crRNAs and the trans-activating small RNA (tracrRNA) were mixed to prepare the guide RNA (gRNA). Male C57BL/6J mice were mated with female BDF1 (C57BL/6 × DBA/2) mice to generate embryos with a hybrid genetic background. At 0.7 d post coitum, the gRNA (30 µmol) and Cas9 protein (1 µg) were injected into the oviductal lumen upstream of the ampulla of pregnant BDF1 females using a glass micropipette. After injection, the oviducts were covered with a piece of Kimwipes paper soaked in saline, grasped in tweezer-type electrodes (Nepa Gene Co., Ltd.), and electroporated using a square-wave pulse generator NEPA21 system (Nepa Gene Co., Ltd.). The electric conditions are as follows: three poring pulses (50 V/5 ms wavelength/50 ms duration/10 decay rate/± polarity) and six transfer pulses (10 V/50 ms wavelength/50 ms duration/40 decay rate/± polarity). After electroporation, the oviduct was returned to its original position in the peritoneal cavity. The epidermis was sutured with clips. The RNA oligos and Cas9 protein were purchased from Integrated DNA Technologies, Inc. The screening of founder pups was performed by PCR and Sanger sequencing. PCR for genotype was performed with the following primers: 5′-GATTCT ATGAGCAGATGAACGG-3′ and 5′-CTTCCAAGGCATGTGGCAAC-3′. Progeny carrying the desired genome edit were backcrossed to C57BL/6J mice. Body weights of mice were recorded weekly from 3 to 9 wk of age. A minimum of 14 mice per genotype and sex were monitored. All mice were housed under standard laboratory conditions with ad libitum access to food and water. All animal experiments were performed in accordance with relevant guidelines and regulations and were approved by the Animal Experiment Committee of Tohoku University (Approval No. 2022MdA-085-04).

## Isolation and cell culture of MEFs

Heterozygous offspring were intercrossed to produce homozygous mutant animals, from which MEFs were isolated as described previously (40, 41). MEFs were cultured in DMEM with 10% FBS and 50 U/ml Penicillin and Streptomycin, 1% MEM-nonessential amino acids, 2 mM L-glutamine, 1 mM sodium pyruvate, and 0.1 mM β-mercaptoethanol (Thermo Fisher Scientific) at 37°C in a humidified 5% $CO_2$ incubator. Early-passage MEFs (passages 2–5) were used for all experiments.

## Preparation of mouse tissue powder for protein and RNA extraction

Mouse tissues, including the brain, heart, liver, skeletal muscle, kidney, spleen, stomach, and small intestine, were rapidly excised and flash-frozen in liquid nitrogen. While still frozen, the tissues were pulverized into fine powder using a pre-chilled mortar and pestle. The resulting tissue powder was used for subsequent total RNA and protein extraction, as described below.

## RNA isolation and reverse transcription (RT)-PCR analysis

Total RNA was extracted from frozen mouse tissue powder and cultured cells with the use of an SV Total RNA Isolation System (Promega) according to the manufacturer's instructions. Reverse transcription (RT) was performed using the PrimeScript RT Reagent Kit (Takara), and the resulting cDNA was used for downstream applications, including conventional PCR followed by gel electrophoresis and Sanger sequencing, or real-time quantitative PCR (qPCR) analysis. qPCR was conducted using a StepOnePlus Real-Time PCR System (Life Technologies) with Fast SYBR Green Master Mix (Life Technologies). Gene expression levels were quantified using the $2^{-\Delta\Delta CT}$ method and normalized to mouse Arbp mRNA. Primer sequences are provided in Table 3.

## Protein extraction and quantification

Frozen tissue powder or cultured cells were lysed on ice for 30 min with Cell Lysis Buffer containing 50 mM Tris–HCl (pH 7.5), 150 mM NaCl, 0.5% NP-40, 10% glycerol, 10 μg/ml aprotinin, 10 μg/ml leupeptin, 1 mM PMSF, 0.4 mM EDTA, 10 mM NaF, 0.4 mM sodium orthovanadate, and 10 mM sodium pyrophosphate. Lysates were centrifuged at 20,000$g$ for 15 min at 4°C, and the resulting supernatant was subjected to protein quantification using Pierce 660 nm Protein Assay Reagent (Thermo Fisher Scientific).

## Lambda phosphatase treatment

Frozen tissue powder was lysed with Cell Lysis Buffer in the absence of phosphatase inhibitors. Then the tissue lysates were treated with Lambda Protein Phosphatase (P0753; New England Biolabs) in 1× NEBuffer Pack for Protein MetalloPhosphatases (50 mM Hepes, 100 mM NaCl, 2 mM DL-Dithiothreitol, 0.01% Brij 35), supplemented with 1 mM $MnCl_2$. The reaction mixture was

**Table 3.  Oligonucleotides used in PCR experiments.**

| Target | Sequence (5′-3′) |
|---|---|
| mEef1d-F (gel) | ATGGCTACAAACTTTCTAGCGCATG |
| mEef1d-R (gel) | CTCCACTTGACGCATAGGAGAGAC |
| mEef1d-F | ATGGAACGCACACAAGTCTC |
| mEef1d-R | CAGAGAAGAAGGCCAAGAAGC |
| mEef1g-F | TGCTGACAGTGACATCGTTC |
| mEef1g-R | GATTCGCTTCACCTCCTCTTTC |
| mEef1b2-F | GTAGAAGACACCACAGGAAGTGG |
| mEef1b2-R | ACTGTGCAAGGCGTTCTTCTCG |
| mVars-F | CCATCAACGGACGTGGCATTCT |
| mVars-R | GGACGGTGATAAAGTAGGCAGG |
| mKtn1-F | TGGAATCCGAGCAAAAGAGGG |
| mKtn1-R | TTCTGCTAGAACTGAAGCGGAG |
| mRrbp1-F | CTCAGCTGGATGAAGCCAAGA |
| mRrbp1-R | TGGCTCCTCATGTCACTCAAC |
| mEef2-F | CAGAAGTACCGTTGTGAGCTGC |
| mEef2-R | GTCAGAGGTTGGCACCATCTTG |
| mArbp-F | GGACCCGAGAAGACCTCCTT |
| mArbp-R | GCACATCACTCAGAATTTCAATGG |

incubated at 37°C for 30 min and was subjected to immunoblot analysis.

## IP using Anti-FLAG M2 magnetic beads

Anti-FLAG M2 magnetic beads (M8823; Sigma-Aldrich) were washed with Wash Buffer (40 mM Tris–HCl [pH 7.5],150 mM NaCl, 0.1% Triton-X-100, 5% Glycerol) and then incubated with cell lysates in Cell Lysis Buffer for 60 min at 4°C with rotation. Beads were washed twice with Wash Buffer and once with Elution Buffer (40 mM Tris–HCl [pH 7.5],150 mM NaCl, 0.05% Triton-X-100). Then, bead-bound proteins were eluted with Elution Buffer supplemented with 500 μg/ml FLAG peptide (Cat#F3290; Sigma-Aldrich) at room temperature for 15 min. Eluates were subjected to mass spectrometry analysis and immunoblot analysis.

## IP using HA-tagged protein purification kit

HA-tagged protein purification kit (#3320; MBL) was used for HA-IP. Briefly, cell lysates were incubated with HA-tag beads in a Spin Column for 60 min at 4°C with rotation. Columns were washed once with Wash Solution (provided in the kit) and eluted with Elution Peptide Solution (provided in the kit). Eluates were subjected to immunoblot analysis.

## IP using Dynabeads Protein G

Dynabeads Protein G (Thermo Fisher Scientific) were washed with Wash Buffer and incubated with the appropriate amount of

antibody (anti-EEF1D or anti-FLAG[M2]) in Wash Buffer for 15 min at room temperature with rotation. Antibody-coated beads were then incubated with cell lysates in Cell Lysis Buffer for 60 min at 4°C with rotation. Beads were washed three times with Wash Buffer, and bead-bound proteins were eluted with 2× SDS sample buffer (diluted from 6× SDS sample buffer: 0.35 M Tris–HCl [pH 6.8], 10% wt/vol SDS, 36% vol/vol Glycerol, 0.012% wt/vol Bromophenol blue, 5% vol/vol beta-mercaptoethanol) and subjected to immunoblot analysis.

### Nascent protein synthesis analysis

C2C12 or Hepa1–6 cells seeded in the 6 cm dishes were treated with the medium containing 10 µg/ml puromycin for 30 min. After removing the supernatant, the cells were directly lysed in the dish and stored. To measure the rate of nascent protein synthesis in vivo, mice were injected intraperitoneally with puromycin at 40 mg/kg body weight. 30 min after the injection, mice were euthanized by cervical dislocation. Tissues were rapidly dissected, immediately snap-frozen in liquid nitrogen, pulverized, and lysed. Cell and tissue lysates were subjected to immunoblot to detect puromycin incorporation.

### Nascent protein synthesis analysis and ER fractionation

C2C12 cells were treated with puromycin for 10 min before cell fractionation. ER and cytoplasmic fractions were isolated from 3 × 10⁷ cells using the Minute ER Enrichment Kit (ER-036; Invent Biotechnologies) according to the manufacturer's instructions. Protein concentrations were determined, and equal amounts of protein were subjected to immunoblotting.

### Gel filtration of C2C12 cell lysates

Cells were grown to ~80–90% confluence before sample collection. Cells were washed with PBS, and Cell Lysis Buffer was added directly to the culture dish. Lysates were incubated on ice for 30 min. The lysates were then clarified by centrifugation at 15,000$g$ for 10 min at 4°C. Protein concentration was determined, and equal amounts of protein from WT and exon 5 KO samples were used for subsequent size exclusion chromatography.

   Size exclusion chromatography was performed on a Superose 6 column (10/300 Gl; Cytiva), at a flow rate of 0.4 ml/min. The column was equilibrated with a buffer composed of 50 mM Tris–HCl (pH 7.5) and 150 mM NaCl. To estimate the molecular size, molecular weight standards were used: thyroglobulin (Sigma-Aldrich), 662 kD; apoferritin (Sigma-Aldrich), 440 kD; β-amylase (Sigma-Aldrich), 200 kD; alcohol dehydrogenase (Sigma-Aldrich), 141 kD; bovine serum albumin (Thermo Fisher Scientific), 67 kD; carbonic anhydrase (Sigma-Aldrich), 29 kD. Each lysate was analyzed separately under identical buffer conditions and was subjected to immunoblotting.

### Immunoblot

Cell lysates, immunoprecipitates, and gel filtration fractions were treated with a final 1× SDS sample buffer (diluted from 6× SDS sample buffer) and heated. Proteins were separated by SDS-

polyacrylamide gel electrophoresis (SDS–PAGE) and transferred onto a polyvinylidene difluoride membrane (IPVH00010; Millipore). The membrane was blocked with 5% wt/vol skimmed milk (Fujifilm Wako) in TBST (Tris Buffered Saline with 0.1% Tween 20) for 30 min at room temperature, then incubated with the appropriate primary antibody at 4°C overnight. After membrane washing with TBST, the membrane was incubated for 20 min at room temperature with the corresponding secondary antibody conjugated to HRP, specific to mouse or rabbit immunoglobulin G (IgG). After membrane washing with TBST, the detection was performed using the ChemiDoc Touch System (Bio-Rad) and enhanced chemiluminescence (ECL) substrates, Pico, Dura, or Femto (Thermo Fisher Scientific). All the antibodies are listed in Table 4.

### Mass spectrometry analysis

For IP-MS analysis of Hepa1-6 cells, bead-bound proteins were eluted with FLAG elution buffer (40 mM Tris–HCl [pH 7.5],150 mM NaCl, 0.05% Triton-X-100, 500 µg/ml FLAG peptide [#F3290; Sigma-Aldrich]), precipitated by the addition of ice-cold 20% trichloroacetic acid. The protein pellet was washed with ice-cold acetone and then digested with trypsin in 100 mM Hepes pH 8.0. The protein digests were purified by using SDB-XC-StageTip (3M). The resulting peptides were injected into a pre-column (L-column micro; CERI), and fractionated on an in-house–fabricated 20-cm column packed with 2-µm octadecyl silane particles (CERI). Elution was performed with a linear gradient of 5–35% solvent B over 50 min at a flow rate of 200 nl/min (solvent A = 0.1% formic acid; solvent B = 0.1% formic acid in acetonitrile) with the use of a Dionex Ultimate 3000 HPLC System (Thermo Fisher Scientific). Eluted peptides were sprayed with a nano-electrospray source and with a column oven set at 42°C (AMR). The Q Exactive Hybrid Quadrupole-Orbitrap mass spectrometer (Thermo Fisher Scientific) was operated in DIA mode. All data were acquired in profile mode with positive polarity. MS1 spectra were collected in the mass/charge (m/z) ratio range of 430 to 860 at a resolution of 35,000 with an automated gain control (AGC) target value of $1 × 10^6$ and a maximum injection time of 50 ms. MS2 spectra were collected in the m/z range of >200 at a resolution of 17,500 with an AGC target value of $1 × 10^6$ and with the automatic maximum ion injection time. Twenty-one DIA windows of 20 units were set from an m/z of 430–850. The normalized collision energy was set to 25%. All DIA raw data were processed with DIA-NN (version 1.8) in the library-free search mode with reference to mouse UniProt sequences (https://www.uniprot.org/. Accessed 27 April 2021). Enzyme specificity was set to "trypsin." Up to zero missed trypsin cleavage was allowed. Normalization was set to "off."

   For IP-MS analysis of C2C12 cells, bead-bound proteins were eluted with FLAG elution buffer (40 mM Tris–HCl [pH 7.5],150 mM NaCl, 0.05% Triton-X-100, 500 µg/ml FLAG peptide [#F3290; Sigma-Aldrich]), precipitated by the addition of ice-cold 20% trichloroacetic acid in the presence of 0.02% sodium deoxycholate. The protein pellet was washed with ice-cold acetone and then digested with trypsin in 100 mM Hepes pH 8.0. The protein digests were purified by using SDB-XC-StageTip (3M). The resulting peptides were injected into a pre-column (L-column2; CERI), and fractionated on an in-house–fabricated 20-cm column packed with 2-µm octadecyl silane particles (CERI). Elution was performed with a

**Table 4. Antibodies.**

| Antibodies | Source | Identifier |
| --- | --- | --- |
| β-actin | Cell Signaling Technology | 3700 |
| Calnexin | Proteintech | 10427-2-AP |
| EEF1D | Proteintech | 10630-1-AP |
| EEF1D | Proteintech | 60085-1-Ig |
| EEF1D | Santa Cruz Biotechnology | SC-393731 |
| KTN1 | Proteintech | 19841-1-AP |
| KTN1 | Santa Cruz Biotechnology | SC-136534 |
| EEF1G | Abcam | ab72368 |
| EEF1G | Santa Cruz Biotechnology | 393378 |
| EEF1B2 | Proteintech | 10483-1-AP |
| GAPDH | Santa Cruz Biotechnology | SC-32233 |
| VARS | Santa Cruz Biotechnology | SC-166674 |
| DDB1 | BD Transduction Laboratories | 612488 |
| RRBP1 | Proteintech | 22015-1-AP |
| VAMP7 | Proteintech | 22268-1-AP |
| HSP70 | BD Transduction Laboratories | 610607 |
| HSP90 | BD Transduction Laboratories | 610418 |
| Anti-FLAG (M2-HRP) | Sigma-Aldrich | A8592 |
| Anti-FLAG (M2) | Sigma-Aldrich | F3165 |
| Anti-HA (3F10-HRP) | Roche | 12013819001 |
| EEF1A1 | Santa Cruz Biotechnology | SC-21758 |
| EEF2 | Proteintech | 20107-1-AP |
| SEC61B | Cell Signaling Technology | 14648 |
| KDEL | Santa Cruz Biotechnology | SC-58774 |
| Puromycin | Millipore | MABE343 |
| Puromycin | Abcam | ab315887 |
| Mouse IgG | Medical & Biological Laboratories | M075-3M2 |
| Rabbit IgG | Sigma-Aldrich | I5006-10 MG |
| HRP-conjugated anti-mouse IgG | Promega | W4021 |
| HRP-conjugated anti-rabbit IgG | Promega | W4011 |
| Goat anti-mouse Alexa488-conjugated antibody | Invitrogen | A11029 |
| Goat anti-rabbit Alexa488-conjugated antibody | Invitrogen | A11008 |
| Goat anti-mouse Alexa594-conjugated antibody | Invitrogen | A11032 |
| Goat anti-rabbit Alexa594-conjugated antibody | Invitrogen | A11037 |

linear gradient of 5–35% solvent B over 50 min at a flow rate of 250 nl/min (solvent A = 0.1% formic acid; solvent B = 0.1% formic acid in acetonitrile) with the use of a Dionex Ultimate 3000 HPLC System (Thermo Fisher Scientific). Eluted peptides were sprayed with a nano-electrospray source and with a column oven set at 42°C (AMR). The Q Exactive Hybrid Quadrupole-Orbitrap mass spectrometer (Thermo Fisher Scientific) was operated in DIA mode. All data were acquired in profile mode with positive polarity. MS1 spectra were collected in the mass/charge (m/z) ratio range of 430–860 at a resolution of 35,000 with an automated gain control (AGC) target value of $1 \times 10^6$ and a maximum injection time of 50 ms. MS2 spectra were collected in the m/z range of >200 at a resolution of 17,500 with an AGC target value of $1 \times 10^6$ and with the automatic maximum ion injection time. Twenty-one DIA windows of 20 m/z units with a 1 m/z overlap were set from m/z 430–850. The normalized collision energy was set to 25%. All DIA raw data were processed with DIA-NN (version 1.8) in the library-free search mode with reference to mouse UniProt sequences (https://

www.uniprot.org/. Accessed 27 April 2021). Enzyme specificity was set to "trypsin." Up to zero missed trypsin cleavage was allowed. Normalization was set to "off."

For whole-proteome analysis, powdered liver tissue and MEFs cultured in dishes were lysed in SDS-containing buffer supplemented with 1% Thermo Halt Protease Inhibitor Cocktail (100×; Thermo Fisher Scientific), followed by incubation at 95°C for 5 min and sonication. The lysis buffer for liver tissue contained 2% SDS and 100 mM Tris–HCl (pH 8.8), whereas that for MEFs contained 1% SDS and 50 mM Tris–HCl (pH 8.8). Before lysis, MEFs were washed once with PBS. After reduction and alkylation, proteins were precipitated with acetone and washed with ice-cold 90% acetone. The resulting protein pellet was digested with trypsin, and the resulting peptide solution was desalted with an EvoTip (Evosep). Peptide fractionation was performed with an EV1109 column (Evosep) and an Evosep One system (Evosep) according to a preprogrammed gradient (60 samples per day) at a flow rate of 1 μl/min. Eluted peptides were sprayed with an emitter (Ref.20-05 LOTUS emitter; AMR) interfaced with an Exploris480 instrument that was equipped with FAIMS Pro. Exploris480 was operated in DIA mode.

Volcano plots were generated using Perseus version 2.0.11.0 (https://maxquant.org/perseus), and statistical significance was assessed using two-tailed $t$ tests. For bar graph representation of a single replicate of Hepa1-6 IP-MS, the raw signal intensity of each protein was normalized to that of EEF1D in the same sample, and the relative enrichment compared with EEF1D was calculated.

### Ribo-seq and RNA-seq analyses

The Ribo-seq library was prepared according to a protocol described previously (42, 43). In brief, ~3 million cells were lysed in 400 μl of a lysis buffer (20 mM Tris–HCl [pH 7.5], 150 mM NaCl, 5 mM MgCl$_2$, 1% Triton X-100, 1 mM dithiothreitol, cycloheximide [100 μg/ml], chloramphenicol [100 μg/ml]), and the lysate was treated with 15 U of Turbo DNase (Thermo Fisher Scientific) and then cleared by centrifugation at 20,000$g$ for 10 min at 4°C. RNA concentration was measured with a Qubit RNA BR Assay (Thermo Fisher Scientific), and lysate containing 10 μg of RNA was digested with 20 U of RNase I (Epicenter) for 45 min at 25°C. The reaction was terminated by the addition of 10 μl of SUPERase·In (Thermo Fisher Scientific), and the digested lysate was subjected to ultracentrifugation at 100,000 rpm for 1 h at 4°C in an Optima MAX-TL centrifuge with a TLA-110 rotor (Beckman Coulter). The RNA in the resulting ribosome pellet was extracted with the use of the Trizol reagent (Thermo Fisher Scientific) and a Direct-zol RNA Microprep Kit (Zymo Research), and RNA fragments of 17–35 nucleotides (nt) were gel-purified after PAGE. The purified RNA fragments were dephosphorylated with T4 polynucleotide kinase (New England Biolabs) and ligated to custom-made preadenylated linkers containing unique molecular identifiers and barcodes for library pooling with the use of T4 RNA ligase 2, truncated KQ (New England Biolabs). The linker-ligated RNA was extracted by PAGE purification, treated with the mouse-rat Ribo-seq riboPOOL (DP-R024-000052; siTOOLs Biotech), and subjected to RT with Protoscript II (New

England Biolabs). The template RNA was hydrolyzed with 1 M NaOH, and the remaining cDNA was extracted by PAGE purification. The purified cDNA was circularized with the use of CircLigaseII ssDNA ligase (Epicenter) and then amplified by PCR with Phusion polymerase (New England Biolabs).

For RNA-seq analysis, RNA was extracted from the same lysate used for Ribo-seq but without RNase treatment. The lysate was mixed with Trizol LS (Thermo Fisher Scientific), and RNA was purified with a Direct-zol RNA Microprep Kit (Zymo Research). The RNA (1 μg) was treated with the NEBNext rRNA Depletion Kit v2 (NEB), and the rRNA-depleted sample was used for RNA-seq library preparation with the NEBNext Ultra II Directional RNA Library Prep Kit for Illumina (NEB).

Libraries for Ribo-seq and RNA-seq were sequenced on a HiSeq X 10 (Illumina) with the 150-nt paired-end read option. Sequence data were processed as previously described with minor modifications (42). In brief, adapter trimming was performed with fastp v.0.21.0, and reads were mapped to noncoding RNA (ncRNA) sequences with the use of STAR v.2.7.0a to remove reads that originated from rRNA genes and other ncRNA genes in silico. The remaining reads were aligned to the mouse genome mm10 and assigned to genes based on the GENCODE mouse vM23 annotation with the use of STAR 2.7.0a. The number of reads after each step is summarized in Table 5.

For quantifying read counts, we used custom scripts (available at https://github.com/ingolia-lab/RiboSeq). A-site offsets of footprints (ribosome-protected fragments, RPFs) were set to 15 nt for most fragment lengths (21–33 nt), except for 23 and 33-nt fragments, for which an offset of 16 nt was used. For RNA-Seq, 15 nt was used for all mRNA fragments.

For metagene analysis around the start codon, the 5′ ends of 30-nt ribosome-foot prints relative to start codons were plotted as mean reads per million (RPM) ± SD from two biological replicates to visualize the 3-nt periodicity. For scaled metagene analysis across transcripts, footprint counts at each codon were quantified along individual transcripts and normalized to RPM. The 5′ UTR, CDS, and 3′ UTR regions of each transcript were proportionally scaled to fixed relative lengths. After scaling, the read distributions were aggregated across transcripts. Transcripts with fewer than one footprint per codon were excluded from the analysis.

Changes in translation efficiency (TE), defined as A-site offset–adjusted ribosome profiling counts normalized by RNA-seq counts, were calculated using DESeq2 (v1.44.0). Reads corresponding to the first and last five codons of coding sequences were excluded from the analysis. Statistical significance was assessed using a likelihood ratio test in a generalized linear model, and significant genes were visualized in an MA plot. For analysis of ER-related gene subsets, transcripts harboring signal peptide sequences (UniProt: KW-0732) were compared with all transcripts annotated in GENCODE mouse vM23. The cumulative distribution of $\log_2$(fold change) was plotted, and statistical significance was determined using the Wilcoxon rank-sum test.

### Immunostaining analysis

Cells were cultured for 24 h on round glass coverslips coated with 0.1% gelatin, washed once with PBS, and fixed for 15 min at

**Table 5.** RNA-seq and Ribo-seq sequencing stats.

| Sample_ID | After filtering and trimming | After ncRNA removal (%) | Uniquely mapped |
|---|---|---|---|
| Ribo-seq (Hepa1-6, WT and Eef1d exon 5 KO #1~3). | | | |
| WT-1 | 35,314,114 | 5,783,890 (16.38) | 3,517,296 |
| KO1-1 | 48,477,307 | 8,307,157 (17.14) | 4,829,841 |
| KO2-1 | 41,034,636 | 6,817,878 (16.61) | 3,998,849 |
| KO3-1 | 40,828,395 | 7,087,984 (17.36) | 4,084,699 |
| WT-2 | 23,575,635 | 3,916,744 (16.61) | 2,379,573 |
| KO1-2 | 12,494,377 | 1,830,167 (14.65) | 1,136,864 |
| KO2-2 | 29,735,922 | 4,656,485 (15.66) | 2,894,502 |
| KO3-2 | 30,609,649 | 4,885,411 (15.96) | 3,109,594 |
| RNA-seq (Hepa1-6, WT and Eef1d exon 5 KO #1~3). | | | |
| RNA-WT-1 | 21,432,156 | 20,312,635 (94.78) | 17,789,402 |
| RNA-KO1-1 | 22,576,747 | 21,237,568 (94.07) | 18,772,682 |
| RNA-KO2-1 | 20,962,712 | 19,671,845 (93.84) | 17,373,718 |
| RNA-KO3-1 | 24,977,664 | 23,601,142 (94.49) | 20,489,641 |
| RNA-WT-2 | 23,247,472 | 21,855,430 (94.01) | 19,229,864 |
| RNA-KO1-2 | 21,164,464 | 19,786,760 (93.49) | 17,550,432 |
| RNA-KO2-2 | 23,931,914 | 22,400,160 (93.60) | 19,887,238 |
| RNA-KO3-2 | 23,989,531 | 22,626,902 (94.32) | 19,719,497 |

Filter and trimming: Number of reads remaining after quality filtering and adapter trimming performed using fastp v0.21.0.
ncRNA removal: Number (and percentage) of reads remaining after removing reads mapped to noncoding RNA (ncRNA) sequences using STAR v2.7.0a.
Uniquely mapped: Number of unique reads aligned to the human genome (mm10) and assigned to the GENCODE mouse vM23 using STAR v2.7.0a.

room temperature with 3.7% PFA in PBS. For puromycin immunostaining experiments, cells were treated with 10 µg/ml puromycin for 30 min before fixation. For experiments requiring WGA staining, fixed cells were washed three times with HBSS, incubated with 5 µg/ml WGA–Alexa 647 (W21404; Thermo Fisher Scientific) in HBSS for 10 min at room temperature, and washed twice with PBS. After fixation (and after WGA labeling when performed), cells were washed once with PBS, permeabilized for 15 min with 0.25% Triton X-100 in PBS, exposed for 30 min at room temperature to blocking solution (5% BSA and 0.25% Triton X-100 in PBS), and incubated overnight at 4°C with primary antibodies in blocking solution. The cells were then washed three times with PBS before incubation for 1 h at room temperature with Alexa488- or Alexa594-conjugated secondary antibodies diluted in PBS containing 0.25% Triton X-100. They were washed twice with PBS, stained for 10 min at room temperature with DAPI (1 µg/ml) in PBS, washed again twice with PBS, and mounted on a glass slide with the use of ProLong Glass Antifade Mountant (Thermo Fisher Scientific). Widefield fluorescence images were acquired using either a BZ-9000 fluorescence microscope (Keyence) equipped with a mercury arc lamp and filter cubes for DAPI, GFP-B (Alexa Fluor 488), and TxRed (Alexa Fluor 594). Images were obtained using 20× or 40× objective lenses. Confocal images were acquired using a laser scanning confocal microscope (LSM780; Zeiss) equipped with a 32-channel spectral detector. Excitation was performed using 405, 488, and 561 nm laser lines, and emission

signals were detected using spectral detection ranges optimized by the Smart Setup (Smartest) function in ZEN software (Zeiss). Images were obtained using 40× or 63× objective lenses with the pinhole set to 1 Airy unit. All the antibodies are listed in Table 4.

For pseudocolor visualization, EEF1D signal images were converted to pseudocolor using the "Fire" lookup table in ImageJ. Pixel intensity values were mapped to a false-color scale to facilitate visual discrimination of differences in signal intensity. In the resulting images, darker colors indicate lower pixel intensities, whereas brighter or hotter colors indicate higher pixel intensities. The accompanying color bar shows the correspondence between pixel intensity values and the pseudocolor scale. Pearson's correlation coefficients were calculated using the Coloc2 plugin in ImageJ (NIH). Regions of interest (ROIs) were defined as rectangular areas encompassing the entire single cell and were analyzed using the ROI Manager tool. Default settings of the Coloc2 plugin were used for all analyses.

### Immunohistochemistry and immunofluorescence analysis of tissues

Age-matched WT and exon 5 KO female mice, aged two to 4 mo, were anesthetized and underwent transcardial perfusion with PBS. Tissues, including the liver and small intestine, were carefully dissected, fixed in Mildform 10N (Cat#133-10311; FUJIFILM Wako Pure

Chemical Corporation) at 4°C for 48 h, and processed for paraffin embedding. Serial 3.5 μm sections were cut and mounted on glass slides. Hematoxylin and eosin staining was performed using standard protocols.

For immunohistochemical detection of EEF1D, the serial sections were deparaffinized, treated with 1× Histofine antigen retrieval solution (pH 9.0; Cat#415211; Nichirei Biosciences) at 120°C for 5 min using an autoclave, and then cooled down in ice-cold water for 40 min. They were then incubated overnight at 4°C with anti-EEF1D antibody (Cat#10630-1-AP; Proteintech) at a 1:400 or 1:200 dilution in PBS containing 0.5% BSA and 0.05% NaN$_3$. Afterward, sections were treated with methanol containing 0.03% H$_2$O$_2$ at room temperature for 20 min, incubated with Histofine Simple Stain Mouse MAX-PO (R) (Cat#414341; Nichirei Biosciences) at room temperature for 40 min, followed by a DAB reaction. The sections were counterstained with hematoxylin and mounted with Multimount 480 (Cat# 0863142211; Matsunami Glass). For immunofluorescence detection of EEF1D and SEC61B, the serial sections were deparaffinized, treated with 1× Histofine antigen retrieval solution (pH 9.0; Cat#415211; Nichirei Biosciences) at 120°C for 5 min using an autoclave, and then cooled down in ice-cold water for 40 min. They were then incubated in PBS containing 5% BSA at room temperature for 60 min and subsequently incubated overnight at 4°C with anti-EEF1D antibody (Cat#10630-1-AP; 1:400; Proteintech) or anti-SEC61B antibody (Cat#14648; 1:500; CST) in PBS containing 0.5% BSA and 0.05% NaN$_3$. Afterward, sections were washed with PBS, incubated with Alexa488- or Alexa594-conjugated secondary antibodies (1:500 dilution) diluted in PBS containing 0.5% BSA and 0.05% NaN$_3$ at room temperature for 2 h. Sections were then washed with PBS and mounted with VECTASHIELD Vibrance Antifade Mounting Medium with DAPI (Cat#H-1800; Vector Laboratories).

### Transmission electron microscopy analysis

For transmission electron microscopy, mice were anesthetized and perfused with PBS followed by 2.5% glutaraldehyde in 0.1 M phosphate buffer (PB) via the portal vein. The small intestine was carefully dissected, fixed overnight at 4°C with 2.5% glutaraldehyde in 0.1 M PB, and stored in 0.1 M PB. The tissue was then postfixed for 2 h at room temperature to 1% OsO$_4$ in 0.1 M PB, dehydrated with a graded series of ethanol solutions, and embedded in Epon 812. Ultrathin sections (70 nm) were collected on copper grids, stained with uranyl acetate and lead citrate, and examined with a transmission electron microscope (JEM-1400Flash; JEOL).

### Hematological and plasma biochemical analysis

For blood testing, peripheral blood was collected into EDTA-coated tubes. White blood cells, RBC, hemoglobin, hematocrit, platelets (Plt), and plateletcrit (Pct) were measured using an automated hematology analyzer for animals (Microsemi LC-662; HORIBA). For biochemical analysis of plasma, peripheral blood in EDTA-coated tubes was centrifuged at 1,100$g$ for 10 min at 4°C, and the supernatant was carefully collected as plasma. Plasma level of

albumin (v-ALB-P, Cat#4547410436389), total protein (v-TP-P, Cat#454741043639), glucose (v-GLU-P, Cat#4547410485578), and triglyceride (v-TG-PIII, Cat#4547410436419) were measured using FUJI DRI-CHEM 7000V analyzer (Fujifilm).

### Statistical analysis

All data are presented as the mean of the independent replicates. Statistical details for each analysis are described in the corresponding Methods sections or figure legends.

## Data Availability

RNA-seq and Ribo-seq data have been deposited in Gene Expression Omnibus (GEO, https://www.ncbi.nlm.nih.gov/geo) under accession number GSE324886. Proteome data have been deposited in Japan ProteOme Standard STandard Repository (jPOSTrepo, https://repository.jpostdb.org) (44) under accession numbers JPST004450 (IP-MS) and JPST004451 (whole proteome). All original code required to reanalyze the data reported in this article has been deposited in Zenodo and is available at reference 45. All unique and stable reagents generated in this study are available from the lead contact with a completed Materials Transfer Agreement.

### Declaration of use of generative AI

A portion of the code used for data processing was developed with assistance from ChatGPT 5.3 (OpenAI). ChatGPT was also used to assist with grammar editing of the manuscript. All AI-generated suggestions were carefully reviewed and modified. The authors take full responsibility for the content of the publication.

## Supplementary Information

## Acknowledgements

This work was supported by the Japan Society for the Promotion of Science (JSPS) KAKENHI (JP20K22616, JP22K06198 to M Hosogane; JP21H02458, JP24K02300 to K Nakayama), by RIKEN (PRI project to S Iwasaki), and by the Support System for Young Researchers for the use of research equipment, instruments, and devices in Tohoku University. We thank Y Nagasawa, M Ando, M Kikuchi, and T Konishi for general technical assistance; our laboratory members for helpful discussions; C Tazawa and N Shibata for technical assistance with immunohistochemistry; H Suda for assistance with the i-GONAD method for mouse generation; Y Sakamaki of the Ochanomizu Research Facility (ORF), Institute of Science Tokyo, for technical assistance with TEM analysis; M Mito of RIKEN for technical assistance with Ribo-seq analysis; the HOKUSAI SailingShip supercomputer facility at RIKEN for providing computational support; and the Biomedical Research Core of Tohoku University Graduate School of Medicine for technical support. Schematic illustrations were generated using BioRender (https://www.biorender.com/).

## Author Contributions

M Jamous: validation, investigation, and writing—original draft.
M Hosogane: conceptualization, data curation, funding acquisition, investigation, writing—original draft, and project administration.
X Huang: investigation and writing—original draft.
M Suzuki: resources.
A Hatano: data curation and formal analysis.
Y Shichino: formal analysis, methodology, and writing—review and editing.
S Iwasaki: conceptualization, supervision, funding acquisition, and writing—review and editing.
K Murayama: methodology.
M Matsumoto: data curation and formal analysis.
K Nakayama: conceptualization, supervision, funding acquisition, and writing—review and editing.

## Conflict of Interest Statement

S Iwasaki is a member of the Scientific Reports editorial board, an associate editor of The Journal of Biochemistry, and a paid consultant of Eisai. Y Shichino is an associate editor of The Journal of Biochemistry. The remaining authors declare that they have no competing interests.

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
