## [Reviewer comments · Life Science Alliance]

The ER Anchoring and Abundance of the EEF1B Complex is Affected by Tissue-Specific EEF1D Splicing

Muhammad Jamous, Masaki Hosogane, Xiaoxin Huang, Mikiko Suzuki, Atsushi Hatano, Yuichi Shichino, Shintaro Iwasaki, Kazutaka Murayama, Masaki Matsumoto, and Keiko Nakayama

DOI: <https://doi.org/10.26508/lsa.202503513>

Corresponding author(s): Keiko Nakayama, Tohoku University and Masaki Hosogane, Dokkyo Medical University

Review Timeline:

Submission Date:	2025-09-20
Editorial Decision:	2025-12-12
Revision Received:	2026-03-31
Editorial Decision:	2026-04-27
Revision Received:	2026-05-11
Accepted:	2026-05-18

Scientific Editor: Sarita Hebbar

Transaction Report:

December 12, 2025

Re: Life Science Alliance manuscript #LSA-2025-03513

Prof. Keiko Nakayama
Tohoku University
Graduate School of Medicine
2-1 Seiryomachi
Aobaku
Sendai, Miyagi 980-8575
Japan

Dear Prof. Nakayama, and Dr. Hosogane,

Thank you for submitting your manuscript entitled "The ER Anchoring and Abundance of EEF1B Complex are Regulated by EEF1D Alternative Splicing" to Life Science Alliance. We apologise for a delay in communicating a decision on your manuscript.

It was assessed by expert reviewers, whose comments are appended to this letter. All three reviewers agree that your description, including the lack of significant phenotypes in mice with exon 5 deletion (EEF1D), will be of potential interest to the field. You will also note that the reviewers have some concerns that preclude publication at this stage.

We encourage you to follow the suggestions of the reviewers to include further characterisation and /or functional data. We leave it optional for you to decide if you will include any of the suggested experiments from the reviewers. In the absence of any new functional data, we suggest that you tone down your conclusion, and modify statements on the regulation of translational machinery by splicing and ER-association.

We agree that you must resolve all the technical points raised by reviewers related to (1) nomenclature and specific terminology used in the study (reviewer 1 and 2), (2) description of method and presentation of results (all reviewers), (3) include relevant citations that are currently missing (reviewer 2).

In line with their overall assessment, we invite you to submit a revised manuscript addressing the reviewers' comments. When submitting the revision, please include a letter addressing the reviewers' comments point by point. While a rebuttal must respond to all points in some form, additional experiments to resolve these points, other than indicated above, will not be required

I would be happy to discuss the revision in more detail via email or phone/videoconferencing. Please let me know which option you prefer, if any.

Thank you for this interesting contribution to Life Science Alliance. We hope that the comments below will prove constructive as your work progresses, and we are looking forward to receiving your revised manuscript.

Sincerely,

Sarita Hebbar, PhD
Scientific Editor
Life Science Alliance
<http://www.lsjournal.org>

B. MANUSCRIPT ORGANIZATION AND FORMATTING:

Reviewer #1 (Comments to the Authors (Required)):

This is an interesting manuscript that ties exon 5 of Eef1d to an association with the protein KTN1, an ER associated protein. The series of experiments seem rather straight forward and the omission of the coding sequence of exon 5 leads to both a loss of interaction with KTN1 while preserving interactions with EEF1G, VARS and EEF1A. Secondly, the KO of exon 5 leads to the reduction in the other EEF1B proteins in liver although there appeared to be no reduction in liver protein synthesis. An equivalent experiment in MEFs did not observe this reduction.

Specific Concerns

1. The authors have chosen to use a common nomenclature for EEF1D although this is not the "formally" agreed upon nomenclature (see Clark et. al, Biochimie 78, 1119-1122, 1996). With this in mind, it would be useful for the authors to insert a table that relates their use of nomenclature to the "standard".
2. Is the function of EEF1D-L (seen in Figure 1A) the same as the other isoforms?
3. The use of the word "regulated" implies the ability to vary concentration or enzyme specific activity. While there is tissue specific expression, the authors have not shown regulation. This reviewer would suggest altering the title to "The ER anchoring and abundance of the EEF1B complex is affected by tissue specific alternate EEF1D splicing".
4. Based upon the values presented in Figure S1B, the relative binding of both RRB1 and KTN1 appear to be 100-fold less than for the binding of the other proteins. Is this still significant? Secondly, Fmr1 association is less in EEF1D depleted proteins and more in EEF1D-M enriched proteins (Figure S1A). How significant is this?
5. The authors describe an ER associated protein "KDEL". However, this is the ER retention signal, not a protein. Is the antibody used specific for this amino acid sequence? If so, this should be more clearly stated.
6. It is known that the aggregates of EEF1B can exist in different degrees of aggregation (molecular weight). Is this true also in the EEF1D knockout and is the distribution the same?

Reviewer #2 (Comments to the Authors (Required)):

This paper describes the effects of deleting exon 5 in eEF1D, a component of the eEF1B complex needed to facilitate GTP exchange on eEF1A during the elongation step of protein synthesis. The spliceform that contains exon 5 seems to mediate anchoring of eEF1B to the endoplasmic reticulum via a direct interaction with KTN1. Surprisingly, mice with homozygous deletion of exon 5 appear to develop normally with no ill-effects.

The introduction is clear and reasonably comprehensive, though quite sparing in use of citations to the literature (this is also the same in the results section, which fails to mention previous work on tissue specific isoforms, though their analysis certainly moves things forward).

A range of techniques and assays systems is used providing good evidence to back up their claims throughout. Specifically they use proteomics to identify changes in the eEF1D interactome with different splice variants- they obviously can't represent different tissues comprehensively but do use two different cell lines . CoIP blots are said to be quantitative (page 8 line 122) but the graphs just bars of a similar size, and no stats. I think this is acceptable for coIP as there's a clear visual difference for RRB1 and KTN1 but would suggest that they at least refer to this as "semi-quantitative". There's generally good evidence for involvement of exon 5, they generate and analyse 3 independently derived clones with the deletion and show well validated, convincing loss of binding.

The change in subcellular localisation looks solid but it seems odd to show results for all three clones with one marker and not with other two, so this might be worth changing in the final version. Presumably the anti-eEF1D antibody detects the exon 5 deleted protein identically? It seems to judging by the input blot in 2F but there's clearly lower expression in the deleted clones overall, so might this affect the IF analysis? I think it looks pretty convincing though, especially as they see the same result with other components of the complex whose expression levels would not presumably be affected.

They then generate 2 lines of mice with an exon 5 deletion, both well validated, with very clear results on westerns followed by analysis in relevant tissues, though it just says "KO" mice- presumably homozygotes? I would have found the initial analysis easier to follow if they had started with the phenotype (which is surprisingly subtle and seems to consist of cellular level changes only, with no observable effect on protein synthesis).

I guess they were hoping for more of a phenotype in the mice, as the normal survival and function argues against a physiological impact of the changes in binding and subcellular localisation in the absence of exon 5, but they are to be commended on carrying out and reporting this additional step. The discussion is well written and pretty comprehensive with reasonable suggestions for follow up work on the KO mice (aging, stressors).

Reviewer #3 (Comments to the Authors (Required)):

Re: Jamous et al., "The ER Anchoring and Abundance of EEF1B Complex are Regulated by EEF1D Alternative Splicing"

This is an investigation of isoforms of the EEF1D/eEF1B δ protein produced by alternative splicing of the EEF1D gene. The authors showed the existence of three major isoforms of the EEF1D protein by Western blotting of phosphatase treated protein extracts. Subsequent RT-PCR analysis determined that the isoforms are products of alternatively spliced transcripts, differing in their inclusion of exons 5 and 6. Mass spectrometry and immunoprecipitation analysis showed that the interaction between the EEF1D protein and key endoplasmic reticulum (ER) proteins such as KTN1 require the inclusion of the EEF1D protein domain encoded by exon 5. Immunohistochemistry and immunofluorescence were used to show that the intracellular distribution of EEF1D is altered by the presence or absence of the protein domain encoded by exon 5. A mouse model harbouring a homozygous knockout of exon 5 of the EEF1D gene is developed. Analysis of the tissue expression of EEF1B complex showed that expression of EEF1B subunits is reduced in the liver in the mouse exon 5 KO model.

The manuscript is of high technical quality. The claims regarding the tissue specific alternative splicing leading to varying levels of EEF1D isoforms; the biochemical analysis of the exon 5 domain required for interaction with the ER protein KTN1; and changes in subcellular distribution of the different isoforms are well supported by experimental data. It is of interest to a wide scientific community, however the functional relevance of the isoforms of EEF1D with respect to protein translation are not thoroughly investigated.

Major points:

The authors conclude that this manuscript "illustrates a previously unrecognized mechanism by which splicing modulates translation machinery through ER association." (lines 35-36) however without further functional validation this is an overstatement of the findings.

The claims that the authors make require substantiation with additional experimental data regarding the functional impact of the exon 5 KO. This is identified by the authors in the discussion, for example, "We have not yet investigated translation efficiency of specific protein subsets particularly secreted proteins." (Lines 311-312).

I feel that the manuscript would be improved by further functional analyses, there are experimental methods which could support the claims:

- Determination that the exon 5 KO specifically influences protein elongation with additional experiments such as ribosome profiling
- Subcellular quantitation of protein translation at the ER in the exon 5 KO versus WT, this could be carried out by puromycin ICC.
- Investigation of any alterations in ER structure or organization in the exon 5 KO model, for example as investigated by Teixeira et al (2025)

Minor points

In addition, there are some points that should be addressed, listed below:

Figure 1A. Could the authors comment on the band seen lower than the 35kDa marker. I cannot see a mention of this band in the text or Figure legend (lines 93-97, 715-719), is this a degradation product?

Figure 1A. Did the authors use protein quantitation or analyse a loading control to equalise the amounts of protein extract analysed for each tissue.

Figure 1F - information should be provided to describe how abundance and fold enrichment was calculated. Is this the same analysis as shown in Fig S1? N numbers are not given in the Figure legend - is this analysis of N = 1?

Figure S1 shows more data, especially regarding other proteins whose expression is altered, but not discussed fully in the text. Could the authors comment?

Fig S2 - methods / technical description of pseudocolor imaging should be included

Fig. 4 Mouse tissue IHC does not give the same resolution regarding the subcellular localisation of EEF1D as the cell culture methodologies. Could the authors explain why immunofluorescence and confocal imaging was not carried out on the tissue sections?

Fig. 5 / In vivo analysis - Why have the authors only analysed the liver tissue? Could the authors compare EEF1D levels in other tissues from the exon 5 KO which show expression of varying amounts of protein isoforms in WT (e.g. heart / muscle) rather than comparison to MEFs.

The impact of EEF1D exon 5 KO on global protein translation has been shown by puromycin incorporation assays (Fig 5, lines 262-267), however in the experimental methods I cannot see addition of harringtonine to inhibit translation initiation. As this is one of the key claims of the paper, that the exon 5 KO reduces EEF1D protein levels, and thereby reducing levels of the EEF1B complex, could it be confirmed whether protein amounts are not impacted at the translation elongation stage by orthogonal analysis, e.g. ribosome profiling?

Response to the reviewer #1

We thank Reviewer #1 for the careful evaluation of our manuscript and for the constructive comments. In response to these comments, we have revised the manuscript and performed additional analyses that we believe have strengthened the study. Our detailed, point-by-point responses are provided below:

The authors have chosen to use a common nomenclature for EEF1D although this is not the “formally” agreed upon nomenclature (see Clark et. al, Biochimie 78, 1119-1122, 1996). With this in mind, it would be useful for the authors to insert a table that relates their use of nomenclature to the “standard”.

We appreciate the reviewer’s comment regarding nomenclature. To ensure consistency with current HGNC and MGI guidelines and with genomic databases, we have adopted approved gene symbols (EEF1D for human and Eef1d for mouse). In the revised manuscript, we clarify at the outset that EEF1D corresponds to the eukaryotic elongation factor 1delta formally referred to as eEF1B δ (**page 5, line 52–53**), and added **new Table 1** summarizing the alternative nomenclature for all components of the EEF1B complex.

Is the function of EEF1D-L (seen in Figure 1A) the same as the other isoforms?

The eEF1B δ L isoform (designated as EEF1D-L in our study) contains a neural and testis-specific alternative exon that encodes a nuclear localization signal, conferring functions beyond its canonical role in translation elongation. EEF1D-L has been shown to act as a transcription factor for genes containing heat-shock elements (HSEs), and mutations affecting this isoform have been associated with intellectual disability in humans. In mice, deletion of the EEF1D-L-specific exon results in audiogenic seizures and fear-associated behavior, indicating an important role for eEF1B δ L in normal brain function, particularly under external stress conditions.

Because EEF1D-L shares the C-terminal region, including the GEF domain, with EEF1D-N, M, and S isoforms, it has been proposed that EEF1D-L may also participate in the formation of the EEF1B complex and contribute to translation elongation. However, overexpression of EEF1D-L has been reported to have negative effects on translation, thus EEF1D-L retains GEF activity toward EEF1A and directly participates in translation remains unclear. We have expanded this explanation in the Introduction (**page 6, line 84–90**) and the Discussion (**page 16, line 399–401**).

The use of the word “regulated” implies the ability to vary concentration or enzyme specific activity. While there is tissue specific expression, the authors have not shown regulation. This reviewer would suggest altering the title to “The ER anchoring and abundance of the EEF1B complex is affected by tissue specific alternate EEF1D splicing”.

We appreciate the reviewer’s feedback regarding the use of the term “regulated.” Our study demonstrates that deletion of exon 5 leads to reduced ER localization and decreased abundance of the EEF1B complex; however, the functional consequences of this isoform-dependent mechanism remain to be fully elucidated. In response to the reviewer’s suggestion, we have revised the title to: “The ER anchoring and abundance of the EEF1B complex is affected by tissue-specific alternative splicing of EEF1D.”

Based upon the values presented in Figure S1B, the relative binding of both RRBP1 and KTN1 appear to be 100-fold less than for the binding of the other proteins. Is this still significant? Secondly, Fmr1 association is less in EEF1D depleted proteins and more in EEF1D-M enriched proteins (Figure S1A). How significant is this?

In the initial submission, we performed a single replicate of IP-MS analysis in two cell lines, C2C12 and Hepa1-6. To identify reliable variant-specific binding partners, we focused on genes commonly identified in both cell lines. This analysis yielded nine candidate proteins including Fmr1, KTN1 and

Data for reviewer 1

RRBP1 as exon 5-dependent interactors (**Data for reviewer, 1A**). However, inspection of the raw MS signal intensities revealed that the signal for Fmr1 was approximately 100-fold lower than that of KTN1, suggesting that Fmr1 may represent a minor binding partner or reflects non-specific enrichment in this experiment (**Data for reviewer, 1B**). Although raw MS signal intensity does not reflect absolute protein abundance and is influenced by peptide ionization efficiency, the consistently low signal raises concerns regarding the robustness of Fmr1 interaction compared with KTN1. To address this issue, during revision, we confirmed additional IP-MS experiments using four biological replicates of C2C12 cells. In these experiments, KTN1 and RRBP1 were reproducibly identified as exon-5 dependent interactors, whereas Fmr1 was not detected (**new Fig 2F**). Furthermore, in our previous study, KTN1-IP-MS independently identified an interaction between KTN1 and EEF1D (**Data for reviewer, 1C**), providing additional support for the specificity of this interaction. Taken together, although this interaction between EEF1D and KTN1 appears weaker than the tight associations observed among the EEF1B complex components, our expanded and replicated analyses support the conclusion that the EEF1D-KTN1 interaction is specific and reproducible.

The authors describe an ER associated protein “KDEL”. However, this is the ER retention signal, not a protein. Is the antibody used specific for this amino acid sequence? If so, this should be more clearly stated.

We appreciate the reviewer’s correction. We agree that it is not accurate to describe “KDEL” as an ER-associated protein. Rather, the anti-KDEL antibody recognizes the ER-retention signal (KDEL motif) present in multiple ER-resident proteins. To clarify this point, we have revised the text in the manuscript to read: “anti-KDEL antibody, which recognizes the ER retention motifs and serves as an ER marker” (**page 10, line 208–209**).

It is known that the aggregates of EEF1B can exist in different degrees of aggregation (molecular weight). Is this true also in the EEF1D knockout and is the distribution the same?

We agree that it is important to assess the extent of aggregates of the EEF1B complex. To address this point, we performed size-exclusion chromatography using a Superose 6 column (Cytiva) to separate protein complexes into fractions spanning low- to high- molecular-weight ranges. This analysis revealed that EEF1B complex components eluted predominantly in high-molecular-weight fractions corresponding to more than 450 kDa, with little to no signal detected in fractions corresponding to monomeric proteins. In contrast, the ~100 kDa monomeric protein EEF2 eluted in the expected low-molecular-weight fractions, confirming appropriate separation and column performance. Importantly, the distribution profiles of EEF1B complex components across high-molecular-weight fractions were comparable between WT and exon 5 KO cells, indicating that deletion of exon 5 does not alter the extent of aggregation of the EEF1B complex. These results

suggest that exon 5 deletion does not disrupt the tight association or assembly state of the EEF1B complex. Because these data provide important insight into the mode of EEF1B complex assembly, we have included this analysis in the revised manuscript as **new Fig 2G** and added the corresponding description (**page 9, line 184–186**).

Response to the reviewer #2

We greatly appreciate the encouraging and constructive comments from the reviewer #2. In response, we have further strengthened the manuscript by conducting additional experiments and clarifying several sections of the text. Our detailed response to the reviewer's comments are provided below:

The introduction is clear and reasonably comprehensive, though quite sparing in use of citations to the literature (this is also the same in the results section, which fails to mention previous work on tissue specific isoforms, though their analysis certainly moves things forward).

In response to the reviewer's comment, we have added appropriate citations throughout the manuscript, particularly in the Introduction and Results sections, to better acknowledge previous work on tissue-specific isoforms and related studies.

CoIP blots are said to be quantitative (page 8 line 122) but the graphs just bars of a similar size, and no stats. I think this is acceptable for coIP as there's a clear visual difference for RRBP1 and KTN1 but would suggest that they at least refer to this as "semi-quantitative".

As suggested by the reviewer, we have revised the wording from "Quantitative" to "semi-quantitative" to describe IP-MS analysis. In the revised manuscripts, we now state "Semi-quantitative comparison of this single IP-MS dataset confirmed exon 5–dependent interactions with RRBP1 and KTN1 (**new supplementary Fig S1C**)" (**page 8, line 143–144**). Although exon 5-dependent interactions with KTN1 were reproducibly observed in two different cell lines, the original IP-MS analysis was performed using a single replicate, which precluded robust statistical evaluation. To address this limitation, we performed additional IP-MS experiments using four biological replicates of C2C12 cells and included statistical analyses. This expanded data set enabled quantitative assessment of interaction changes based on log₂ fold enrichment and -log₁₀ p value calculated using student t-test.

The change in subcellular localisation looks solid but it seems odd to show results for all three clones with one marker and not with other two, so this might be worth changing in the final version.

We thank the reviewer for this helpful suggestion. In the revised manuscript, we now include representative immunofluorescence images for all analyzed clones. Specifically, images for all clones are shown in **new Fig 3C** for co-immunostaining of EEF1G and KDEL, and in **new**

supplementary Fig S2B for co-immunostaining of EEF1D and KDEL. We believe that this revision allows readers to more readily associate the representative images with the corresponding quantitative analyses.

Presumably the anti-eEF1D antibody detects the exon 5 deleted protein identically? It seems to judging by the input blot in 2F but there's clearly lower expression in the deleted clones overall, so might this affect the IF analysis? I think it looks pretty convincing though, especially as they see the same result with other components of the complex whose expression levels would not presumably be affected.

We thank the reviewer for raising this important point. According to information provided by the antibody manufacturers (Santa Cruz and Proteintech), the epitopes used to generate the anti-EEF1D antibodies spans a broad region of the EEF1D protein, from the N-terminus to C-terminus, and includes the region encoded by exon 5. To assess the specificity of these antibodies toward different EEF1D isoforms, we performed immunoblotting using C2C12 cells overexpressing FLAG-tagged EEF1D-M, -N, or -S, and compared the signal intensities obtained by using anti-FLAG and anti-EEF1D antibodies (**Data for reviewer 2**). Under these conditions, both Proteintech and Santa Cruz anti-EEF1D antibodies detected EEF1D-S with signal intensities similar to, or slightly higher than, those of EEF1D-N, when we compared with the signal intensity using an anti-FLAG antibody.

Data for reviewer 2

Based on these results, we cannot completely exclude the possibility that deletion of exon 5 affects antibody reactivity toward the exon 5-deleted EEF1D protein in immunostaining experiments, although our analyses were performed using two independent EEF1D antibodies (**new Fig 3A and supplementary Fig S2**). Nevertheless, even if antibody reactivity affects signal intensity, the subcellular localization of EEF1D can still be reliably evaluated. Importantly, we observed consistent changes in subcellular localization not only for EEF1D but also for its binding partner EEF1G, whose antibody reactivity is not expected to be affected by exon 5 deletion. Together, these

results support the conclusion that exon 5 deletion alters ER association of the EEF1B complex.

They then generate 2 lines of mice with an exon 5 deletion, both well validated, with very clear results on westerns followed by analysis in relevant tissues, though it just says “KO” mice—presumably homozygotes?

We thank the reviewer for this important clarification. Unless otherwise specified, all experiments were performed using homozygous exon 5 KO. To avoid ambiguity, we have added the following statement to the Results section of the revised manuscripts; “All subsequent analyses were performed using homozygous exon 5 KO mice line #1 unless otherwise specified” (**page 11, line 262–263**).

I would have found the initial analysis easier to follow if they had started with the phenotype (which is surprisingly subtle and seems to consist of cellular level changes only, with no observable effect on protein synthesis).

We thank the reviewer for this thoughtful suggestion. To improve the overall readability and logical flow of the manuscript, we have reorganized the Results section by presenting the phenotypic analysis prior to the detailed examination of EEF1D localization in tissues and MEFs. Specifically, the phenotypic analysis has been moved to **new Fig 5**, allowing readers to first appreciate the overall impact of exon 5 deletion before considering the underlying cellular and molecular mechanisms. We believe that this reorganization improves clarity and better guides the reader through the study.

Response to the reviewer#3

We thank Reviewer #3 for the careful and thorough evaluation of our manuscript. We appreciate the reviewer’s recognition of the significance of our findings, as well as the thoughtful comments highlighting need for further functional validation. We have taken the suggestions seriously and have conducted additional functional analyses, as detailed below.

The authors conclude that this manuscript “illustrates a previously unrecognized mechanism by which splicing modulates translation machinery through ER association.” (lines 35-36) however without further functional validation this is an overstatement of the findings.

The claims that the authors make require substantiation with additional experimental data regarding the functional impact of the exon 5 KO. This is identified by the authors in the discussion, for example, “We have not yet investigated translation efficiency of specific protein subsets particularly secreted proteins.” (Lines 311-312).

I feel that the manuscript would be improved by further functional analyses, there are experimental methods which could support the claims:

- Determination that the exon 5 KO specifically influences protein elongation with additional experiments such as ribosome profiling
- Subcellular quantitation of protein translation at the ER in the exon 5 KO versus WT, this could be carried out by puromycin ICC.
- Investigation of any alterations in ER structure or organization in the exon 5 KO model, for example as investigated by Teixeira et al (2025)

We fully acknowledge the reviewer's concern that our initial submission lacked sufficient functional validation. To address this issue, we performed ribo-seq (**new Figs 4D-4G**), puromycin ICC (**new supplementary Figs S3C and S3D**), and transmission electron microscopy (TEM) of the ER (**new supplementary Fig S6A**), as suggested by reviewer. For ribo-seq, we used WT and exon 5 KO Hepal-6 cells. Ribo-seq revealed no substantial difference in global translation, suggesting that ER-localization of EEF1D is dispensable for overall protein synthesis. Puromycin ICC and immunoblot using ER fractions from WT and exon 5 KO C2C12 cells likewise showed no detectable difference, indicating that ER-associated translation was not compromised by deletion of exon 5. TEM analysis showed typical ER organization in small intestine from both WT and exon 5 KO mice, distinct from the ER aggregation reported by Teixeira et al (2025). Collectively, these new functional analyses provide a more precise characterization of the effects of exon 5 deletion on translation and ER organization.

Data for reviewer 3

A

B

C

Figure 1A. Could the authors comment on the band seen lower than the 35kDa marker. I cannot see a mention of this band in the text or Figure legend (lines 93-97, 715-719), is this a degradation product?

We examined whether an alternative isoform could account for the band detected below the 35 kDa marker. In the GENCODE VM23 Comprehensive Transcript set, we identified a candidate transcript (ENSMUST00000151066.7) encoding an *Eef1d* isoform lacking exons 7 and 8 in addition to exon 5 and 6 (**Data for reviewer, 3A**). Because this isoform could explain the band migrating below EEF1D-S, we performed RT-PCR to assess its expression in mouse tissues. Our primer set was expected to yield a 144-bp products (**Data for reviewer, 3B**), but we did not detect this band in liver and muscle from the sample used in **new Fig 1B**, nor in an independent replicate including additional tissues (**Data for reviewer, 3C**). These results suggest that ENSMUST00000151066.7 if expressed at all is present at levels far lower than the transcripts encoding EEF1D-M, N and S and therefore does not explain the comparable immunoblot to EEF1D-S.

To further evaluate the protein band, we tested three independent EEF1D antibodies on four replicate WT mouse liver lysate. The antibody used in Fig. 1A (Proteintech,10630-1-AP) detected the sub-35-kDa band in three of four replicates (**new supplementary Fig S1A, top**), whereas the other two antibodies (SantaCruz, SC-393731; Proteintech, 60085-1-IG) did not detect this band in any replicate (**new supplementary Fig S1A, bottom**). These data indicate that the sub-35kDa band is likely a nonspecific signal or a degradation product recognized only by Proteintech,10630-1-AP. We have added this note on antibody specificity to revised manuscript (**page 7, line 115–116**).

Figure 1A. Did the authors use protein quantitation or analyse a loading control to equalise the amounts of protein extract analysed for each tissue.

We thank the reviewer for raising this important point. For the experiment shown in **Fig 1A**, total protein concentrations were first quantified using a standard assay. Because endogenous EEF1D expression levels vary substantially across tissues, equal total protein loading resulted in under-representation of low-abundance isoforms in certain samples (**Data for reviewer, 4A**). To visualize isoform patterns more clearly, we adjusted the amount of lysate loaded so as to achieve comparable EEF1D signal intensity across tissues (**Data for reviewer, 4B**), as the aim of **Fig 1A** was to compare isoform composition rather than total abundance, we have clarified this in the revised figure legend of **new Fig 1A** as “Protein amounts loaded for the blot were adjusted to achieve comparable EEF1D signal intensity across tissues.”.

A

*Optimization experiment.
The same amounts of lysate for SDS-PAGE,
after quantitation of protein concentration
using Pierce 660nm Protein Assay Reagent*

B

*Optimization experiment.
Determination of comparable EEF1D signal by
adjusting the amounts of lysate for SDS-PAGE*

Figure 1F - information should be provided to describe how abundance and fold enrichment was calculated. Is this the same analysis as shown in Fig S1? N numbers are not given in the Figure legend - is this analysis of N = 1?

In the revised manuscript, we performed additional IP-MS analyses with four biological replicates in C2C12 cells and conducted statistical testing. Volcano plots in the **new Figs 1E, 1F, 1G, and supplementary Fig S1B** were generated in Perseus version 2.0.11.0 (<https://maxquant.net/perseus>), displaying $\log_2(\text{fold change})$ on the x-axis and $-\log_{10}(\text{p value})$ from two-tailed Student's *t*-test on the y-axis. Because the key findings regarding the exon 5-dependent interaction with KTN1 from the initial submission was reproduced with improved data quality, we have replaced the original single-replicate C2C12 dataset with these new data. The single-replicate Hepa1-6 dataset from the initial submission is now presented separately as **new supplementary Fig S1C**, with the replicate number clearly indicated ($n = 1$) in both the panel and the legend, and with details of data representation provided in the Materials and Methods section.

Figure S1 shows more data, especially regarding other proteins whose expression is altered, but not discussed fully in the text. Could the authors comment?

Based on our new IP-MS data, in addition to KTN1 and RRBP1, we identified VAMP7 and other SNARE-family proteins as significantly and differentially enriched binders of the EEF1D variants. We discussed these genes in the revised Discussion (**page 17, line 417–426**). For access to the raw data and the complete protein lists, we have deposited our proteomics datasets in the Japan ProteOme STandard Repository (jPOSTrepo, <https://repository.jpostdb.org>) under accession numbers JPST004450 (IP-MS) and JPST004451 (whole peoteome),

Fig S2 - methods / technical description of pseudocolor imaging should be included

We placed the detailed description of pseudocolor in the materials and methods section (**page 29, line 784–789**).

Fig. 4 Mouse tissue IHC does not give the same resolution regarding the subcellular localisation of EEF1D as the cell culture methodologies. Could the authors explain why immunofluorescence and confocal imaging was not carried out on the tissue sections?

We thank the reviewer for this constructive comment. To improve spatial resolution, we agree that immunofluorescence and confocal imaging are appropriate for tissue analysis. Accordingly, we stained small intestine and liver sections with an anti-EEF1D antibody and an Alexa Fluor 594-conjugated secondary antibody, and acquired images using a confocal microscope (**new Figs 5D and 5E, bottom**). This analysis provided resolution comparable to our cell-culture experiments, and recapitulated exon 5-dependent localization of EEF1D shown in the original histochemical data.

Furthermore, we performed immunofluorescence with an anti-SEC61B antibody to delineate the ER (new Figs 5D and 5E, top). This staining revealed the ER network in hepatocyte and showed stronger signal at the apical/luminal side of small intestinal cells, facilitating comparison of EEF1D and ER localization in WT and exon 5 KO tissues.

Fig. 5 / In vivo analysis - Why have the authors only analysed the liver tissue? Could the authors compare EEF1D levels in other tissues from the exon 5 KO which show expression of varying amounts of protein isoforms in WT (e.g. heart / muscle) rather than comparison to MEFs.

We appreciate the reviewer's insightful comment. To extend the observation in exon 5 KO liver to additional tissues, we performed immunoblotting of heart, skeletal muscle, small intestine, and brain (new Figs 6B and 6C). Interestingly, heart and skeletal muscle from exon 5 KO mice showed decreased level of EEF1B complex components, whereas small intestine and brain did not exhibit detectable changes. These results suggest tissue type-specific regulation of EEF1B component abundance in response to exon 5 deletion. We have incorporated this point into the revised manuscript (page 13, lines 317–327).

The impact of EEF1D exon 5 KO on global protein translation has been shown by puromycin incorporation assays (Fig 5, lines 262-267), however in the experimental methods I cannot see addition of harringtonine to inhibit translation initiation. As this is one of the key claims of the paper, that the exon 5 KO reduces EEF1D protein levels, and thereby reducing levels of the EEF1B complex, could it be confirmed whether protein amounts are not impacted at the translation elongation stage by orthogonal analysis, e.g. ribosome profiling?

We thank the reviewer for highlighting this important point. Following the reviewer's suggestion, we performed ribo-seq analysis using WT and exon 5 KO Hepal-6 cells. If translation elongation is substantially impaired, ribosomal-protected fragments (RPFs) tend to accumulate toward the 5' end of the coding sequence (CDS) due to reduced ribosome transit. Our metagene analysis revealed no such 5'-end bias in the exon 5 KO cells (new Fig 4E), suggesting that exon 5 deletion has a limited effect on elongation. We further evaluated translation efficiency (TE) by integrating ribo-seq with RNA-seq, and found no major difference in either global translation or the ER-associated translation (new Figs 4F and 4G). Taken together, at least in vitro, these data indicate that dissociation of EEF1D from the ER did not substantially impact translation elongation.

April 27, 2026

RE: Life Science Alliance Manuscript #LSA-2025-03513R

Prof. Keiko Nakayama
Tohoku University
Graduate School of Medicine
2-1 Seiryomachi
Aobaku
Sendai, Miyagi 980-8575
Japan

Dear Prof. Nakayama,

Thank you for submitting your revised manuscript entitled "The ER Anchoring and Abundance of the EEF1B Complex is Affected by Tissue-Specific EEF1D Splicing". We apologise for the delay in communicating our decision due to editor availability issues.

Your revised manuscript was reviewed by two of the original reviewers whose comments are appended below.

As you will read, the two reviewers commented that the revised manuscript has largely addressed their previous concerns. However Reviewer 1 noted some concerns that are still pending, and we agree that you must resolve them with appropriate edits to your manuscript. These are minimal edits and so we hope they will not take you much time:

-overstatement of results (requirement of exon 5 versus its potential to influence or enhance the EEF1D association with SEC1B (concern 1) and ER localisation (concern 3).

-confounding data, (1) minimal change in cell growth in 4A that is unexpected, and (2) difference in dependence of exon 5 in pulldowns in Figure 2 versus 3B/D, that need to be explained and resolved in the manuscript.

-lack of information on ratio of soluble EEF1D to membrane bound EEF1D and molar ratio of KTN1 to EEF1D in the cell to accurately describe the loss of EEF1D when exon 5 is deleted.

We would be happy to publish your paper in Life Science Alliance pending resolution of the reviewer's concerns and final changes necessary to meet our formatting guidelines.

MANUSCRIPT ORGANIZATION AND FORMATTING:

To avoid unnecessary delays in the acceptance and publication of your paper, please read the following information carefully. Full guidelines are available on our Instructions for Authors page, <https://www.life-science-alliance.org/authors>

-For reporting data from experiments on mice, please confirm that all experiments were performed in accordance with relevant guidelines and regulations and include a statement in the Materials and Methods identifying the institutional and/or licensing committee approving the experiments.

-Thank you for providing a data availability statement. Please confirm if "accession number JPST004450 (IP-MS)" includes the data from all sets of describe IP-MS experiments (Hepa1-6, C2C12 cells)

-Please provide a citation or provide more details for these sentences in the methods,

"The CDS of human EEF1D with exon 5 inclusion (EEF1D-N), exon 5 exclusion (EEF1D-S), and partial leucine zipper deletion (EEF1D-N(Δ LZ)) were obtained from plasmids pF1KE2837, pF1KE2835 and pF1KE2836, respectively (Kazusa clone)"

"The CDS of human KTN1 WT lacking a stop codon was obtained from pF1KE1030 (Kazusa clone)."

-Please provide information on fluorescence imaging (excitation/emission filters, objective details).

-Please upload your main and supplementary figures as single files.

-Please remind your secondary Corresponding Author to connect their account with an Orcid ID. They should have received instructions on this.

-Please add the X and Bluesky handles of your host institute/organization, as well as your own, and/or one of the authors, in our system.

-Please remove Graphical Abstract from the manuscript file and upload it as a separate file with the file designation "Graphical Abstract".

-The contributions selected for Shintaro Iwasaki do not qualify them for authorship. Please either update the contributions in our system and in the Author Contributions section of the manuscript, or let us know if the author needs to be removed (and added eventually to the acknowledgment section).

-Please use the [10 author names et al.] format in your references (i.e., limit the author names to the first 10).

-Please be sure that the authorship listing and order is correct.

We welcome submissions of potential cover images for the issue of LSA in which your work would appear. If you have high quality images associated with this work, please feel free to email these, with a caption, to the journal office.

LSA encourages authors to provide a 30-60 second video where the study is briefly explained. We will use these videos on social media to promote the published paper and the presenting author (for examples, see <https://docs.google.com/document/d/1-UWCfbE4pGcDdcgzcmiuJI2XMBJnxKYeqRvLLrLSo8s/edit?usp=sharing>). Corresponding or first-authors are welcome to submit the video. Please submit only one video per manuscript. The video can be emailed to contact@life-science-alliance.org

FINAL FILES:

The following items are required for acceptance.

The license to publish form must be signed before your manuscript can be sent to production. A link to the license to publish form will be available to the corresponding author only. Please take a moment to check your funder requirements.

Thank you for your attention to these final processing requirements. Please revise and format the manuscript and upload materials as soon as you are able.

Thank you for this interesting contribution to the literature. We look forward to publishing your paper in Life Science Alliance.

Sincerely,

Sarita Hebbar, PhD
Scientific Editor
Life Science Alliance
<http://www.lsjournal.org>

Reviewer #1 (Comments to the Authors (Required)):

This revised manuscript is now viewed as generally acceptable although a few concerns still linger. The most notable one is what is the ratio of soluble EEF1D to membrane bound EEF1D and what is the molar ratio of KTN1 to EEF1D in the cell? This is

important in making attempts to accurately quantitate the significance of the loss of EEF1D when exon 5 is deleted and its global effect on protein synthesis.

Specific Concerns

1. Figure 3B appears to show about a 30% loss of EEF1D associated with SEC1B. Thus, exon 5 would not seem to be an absolute requirement for ER association but an enhancement. The authors' comments do not seem to reflect this (i.e. line 222 "...is mediated through the -helical domain").
2. Line 227 - "...with a significant reduction ..." While there is a "statistically significant reduction", in the usual English usage a significant reduction would mean something in the 30 to 50% range. It is also noted that the change in cell growth noted in Figure 4A may be more pronounced, but it is still rather minimal.
3. Line 302 - "Together these results indicate that exon 5 is required for proper ER-associated localization...". This reviewer would argue that "required" overstates the case and "enhances" might be a better choice of words.
4. Figure 2 shows a clear dependence of exon 5 to pull down KTN1. However, Figure 3B and 3D do not show this same complete dependence. Is there an explanation for this difference?

Reviewer #2 (Comments to the Authors (Required)):

The authors have responded well to criticisms and I am happy that the paper is now appropriate for publication.

Response to the reviewer #1

We are grateful to Reviewer #1 for the careful assessment of our manuscript and for the constructive comments. In response, we have revised the manuscript and modified the wording to better reflect the observed results. Detailed, point-by-point responses are provided below.

The most notable one is what is the ratio of soluble EEF1D to membrane bound EEF1D and what is the molar ratio of KTN1 to EEF1D in the cell? This is important in making attempts to accurately quantitate the significance of the loss of EEF1D when exon 5 is deleted and its global effect on protein synthesis.

We thank the reviewer for this important point. To provide additional quantitative insight, we performed densitometric analysis of EEF1D isoforms using three independent antibodies, based on the data originally presented in Supplementary Fig. S1A. Across these analyses, the relative ratios of EEF1D-S to EEF1D-N in mouse liver were estimated to be approximately 1:0.69, 1:0.45, and 1:0.42, respectively. These values provide a rough estimate of relative EEF1D abundance; however, they do not distinguish the fraction of EEF1D-N associated with the ER membrane and therefore do not provide a direct measure of ER membrane-bound versus soluble pools. Regarding the molar ratio between KTN1 and EEF1D, we did not attempt direct quantification, as this would require accurate cross-comparison between different antibodies, which is inherently challenging due to differences in antibody affinity and detection efficiency. Absolute quantification using mass spectrometry in combination with appropriate standards would be required to determine this ratio reliably.

We note that the input lanes in Fig. 2F correspond to 2% of total lysate used for immunoprecipitation, and IP/input values were estimated from band intensities to allow semi-quantitative comparison of protein recovery. As summarized in the table for the reviewer #1, semi-quantitative analysis of the EEF1D immunoprecipitation experiment showed recovery of KTN1 together with EEF1D, with estimated IP/input values of approximately 1.2% for KTN1 (Proteintech) and 0.65% for KTN1 (SantaCruz). Based on these values, the fraction of KTN1 bound to EEF1D was estimated to be approximately 15–30%. This value was comparable to that of EEF1A1 (~18%) under the same conditions, but lower than that of EEF1G (~100%) and VARS (~260%). These results suggest that a substantial subset of total cellular KTN1 is associated with EEF1D under these conditions. These comparisons should be interpreted with caution, as differences in antibody sensitivity, protein abundance, and detection efficiency may influence the apparent recovery.

Table for the reviewer #1	EEF1D (Bait)	KTN1 (Proteintech)	KTN1 (SantaCruz)	EEF1A1	VARs	EEF1G	HSP90 (negative ctrl)
Input (2%) band intensity quantification	3.532E+06	1.096E+07	9.713E+06	1.069E+07	9.885E+05	5.014E+06	1.270E+07
IP band intensity quantification	7.209E+06	6.770E+06	3.163E+06	3.870E+06	5.285E+06	1.091E+07	5.474E+04
IP efficiency (%)	4.08	1.24	0.65	0.72	10.69	4.35	0.01
Estimated fraction bound to EEF1D (%)	100	30.26	15.9	17.74	261.94	106.59	0.21

We have included the densitometric analysis of EEF1D isoforms using three independent antibodies in **the revised Supplementary Fig. S1A**. Because this analysis was performed using four biological replicates, we considered it appropriate to include these data in the revised manuscript. In contrast, the immunoprecipitation-based comparison of protein recovery was derived from a representative experiment and is therefore provided here only as a semi-quantitative reference for interpretation. We have clarified these limitations in the Discussion (**New Lines 425–435**).

Figure 3B appears to show about a 30% loss of EEF1D associated with SEC1B. Thus, exon 5 would not seem to be an absolute requirement for ER association but an enhancement. The authors' comments do not seem to reflect this (i.e. line 222 "...is mediated through the α -helical domain").

We thank the reviewer for this insightful comment. We agree that the reduction in EEF1D association observed in Figure 3B does not support a model in which exon 5 is absolutely required for ER association, but rather suggests that exon 5 contributes to this process. We have therefore revised the wording to better reflect this interpretation. Specifically, we have modified the description from “is mediated through the α -helical domain” to “is largely mediated through the α -helical domain” (**New Line 220**).

We would also like to clarify that Pearson’s correlation coefficients in our analysis do not directly represent the fraction of protein associated with the ER, but rather reflect the degree of spatial overlap between signals within the selected region of interest (ROI). In our analysis, ROIs were defined as rectangular areas encompassing the entire cell. As a result, the values primarily reflect overall spatial overlap rather than strictly compartment-specific association, which may limit sensitivity to detect compartment-specific changes in ER association. Therefore, a decrease from ~0.9 to ~0.6 should not be interpreted as a simple proportional reduction in ER-associated EEF1D, but instead indicates a clear redistribution of EEF1D within the cell. To facilitate interpretation of the colocalization analysis, we have clarified the definition of ROIs in the Methods section (**New Lines 806–809**).

Line 227 - "...with a significant reduction ..." While there is a "statistically significant reduction", in the usual English usage a significant reduction would mean something in the 30 to 50% range. It is also noted that the change in cell growth noted in Figure 4A may be more pronounced, but it is still rather minimal.

We thank the reviewer for this important comment. We agree that the term “significant reduction” may imply a larger effect size than what is observed in our data. Similarly, the change in cell growth observed in Figure 4A is detectable but limited in magnitude. We have therefore modified the text to more accurately reflect the extent of the observed effects, as below.

New Line 225: "with a modest but statistically significant reduction".

New Line 234: “Compared with C2C12 cells, exon 5 KO Hepa1-6 cells showed a slightly greater reduction in cell growth, whereas nascent protein synthesis remained unchanged as assessed by puromycin incorporation assays (Figs 4A and 4B).”

Line 302 - "Together these results indicate that exon 5 is required for proper ER-associated localization...". This reviewer would argue that "required" overstates the case and "enhances" might be a better choice of words.

We thank the reviewer for this insightful comment. We agree that the term “required” may overstate the extent of dependence. We have therefore revised the wording to “enhances” to more accurately reflect the contribution of exon 5 to ER-associated localization.

New Line 300: “Together, these results indicate that exon 5 enhances ER-associated localization of EEF1D in mouse tissues.”

Figure 2 shows a clear dependence of exon 5 to pull down KTN1. However, Figure 3B and 3D do not show this same complete dependence. Is there an explanation for this difference?

We thank the reviewer for this important point. While Figure 2 indicates a strong dependence of exon 5 for KTN1 pull-down, the Pearson’s correlation coefficients in Figure 3 do not reach complete separation. This difference likely reflects both technical and biological factors. From a technical perspective, the dynamic range of Pearson’s coefficients is constrained by imaging resolution, background signal, and ROI-based analysis, which can prevent values from reaching theoretical extremes (i.e., 1 or 0). Therefore, a decrease from ~0.9 to ~0.6 still represents a clear reduction in colocalization. From a biological perspective, it is also possible that EEF1D localization is not exclusively dependent on KTN1, and that other factors contribute to its recruitment to the ER. In this context, exon 5 may function to enhance or stabilize the association with KTN1 rather than being absolutely required for localization. Taken together, our data support that exon 5 is an important determinant that enhances KTN1-associated localization, while residual spatial overlap may reflect both technical limitations and additional biological mechanisms.

Response to the editor

For reporting data from experiments on mice, please confirm that all experiments were performed in accordance with relevant guidelines and regulations and include a statement in the Materials and Methods identifying the institutional and/or licensing committee approving the experiments.

We thank the editor for this important point. We confirm that all animal experiments were performed in accordance with relevant guidelines and regulations and were approved by the Animal Experiment Committee of Tohoku University (Approval No. 2022MdA-085-04). We have added this information to the Materials and Methods section (**New Lines 534–536**).

Please confirm if "accession number JPST004450 (IP-MS)" includes the data from all sets of describe IP-MS experiments (Hepa1-6, C2C12 cells).

We confirm that the accession number JPST004450 includes both Hepa1-6 and C2C12 cells data.

Please provide a citation or provide more details for these sentences in the methods (about source of plasmids).

We have clarified the source of the plasmids obtained from the Kazusa DNA Research Institute, replaced the plasmid names with the corresponding product IDs from the Kazusa cDNA clone collection, and added appropriate descriptions in the Methods section (**New Lines 468–471, and 477–478**).

Please provide information on fluorescence imaging (excitation/emission filters, objective details).

We have expanded the Methods section to include detailed information on fluorescence imaging conditions, including excitation sources, emission detection settings, and objective lenses (**New Lines 792–800**).

Please use the [10 author names et al.] format in your references (i.e., limit the author names to the first 10).

We have revised the reference format using the Life Science Alliance (Citation Sequence) style in EndNote X9.

Additional corrections

We identified errors in **the antibody names in the figure legends of Figures 3A, S2B, and S5C**, which have now been corrected in the revised manuscript. In addition, we found an error in **the labeling of Figure 6C** during the revision process. The labels for the rightmost panels (Small intestine and Brain) were inadvertently reversed and have now been corrected. These errors do not affect the

interpretation of the results. We apologize for these mistakes.

May 18, 2026

RE: Life Science Alliance Manuscript #LSA-2025-03513RR

Prof. Keiko Nakayama
Tohoku University
Graduate School of Medicine
2-1 Seiryomachi
Aobaku
Sendai, Miyagi 980-8575
Japan

Dear Dr. Nakayama,

Thank you for submitting your Research Article entitled "The ER Anchoring and Abundance of the EEF1B Complex is Affected by Tissue-Specific EEF1D Splicing". It is a pleasure to let you know that your manuscript is now accepted for publication in Life Science Alliance. Congratulations on this interesting work.

Your article will publish open access upon publication under a CC-BY license.

DISTRIBUTION OF MATERIALS:

Again, congratulations on a very nice paper. I hope you found the review process to be constructive and are pleased with how the manuscript was handled editorially. We look forward to future exciting submissions from your lab.

Sincerely,

Sarita Hebbar, PhD
Scientific Editor
Life Science Alliance
<http://www.lsajournal.org>